# Structural basis for SARM1 inhibition and activation under energetic stress

**Michael Sporny[1†], Julia Guez-Haddad[1†], Tami Khazma[1†], Avraham Yaron[2], Moshe Dessau[3], Yoel Shkolnisky[4], Carsten Mim[5‡], Michail N Isupov[6‡], Ran Zalk[7‡], Michael Hons[8‡], Yarden Opatowsky[1]\***

[1]The Mina & Everard Goodman Faculty of Life Sciences, Bar-Ilan University, Ramat-Gan, Israel; [2]Department of Biomolecular Sciences, Weizmann Institute of Science, Rehovot, Israel; [3]Azrieli Faculty of Medicine, Bar Ilan University, Safed, Israel; [4]Department of Applied Mathematics, School of Mathematical Sciences, Tel-Aviv University, Tel-Aviv, Israel; [5]Royal Technical Institute (KTH), Dept. For Biomedical Engineering and Health Systems, Stockholm, Sweden; [6]Biosciences, University of Exeter, Exeter, United Kingdom; [7]National Institute for Biotechnology in the Negev, Ben-Gurion University of the Negev, Beer-Sheva, Israel; [8]European Molecular Biology Laboratory, Grenoble, France

**\*For correspondence:**
yarden.opatowsky@biu.ac.il

[†]These authors contributed equally to this work
[‡]These authors also contributed equally to this work

**Competing interests:** The authors declare that no competing interests exist.

**Abstract** SARM1, an executor of axonal degeneration, displays NADase activity that depletes the key cellular metabolite, NAD+, in response to nerve injury. The basis of SARM1 inhibition and its activation under stress conditions are still unknown. Here, we present cryo-EM maps of SARM1 at 2.9 and 2.7 Å resolutions. These indicate that SARM1 homo-octamer avoids premature activation by assuming a packed conformation, with ordered inner and peripheral rings, that prevents dimerization and activation of the catalytic domains. This inactive conformation is stabilized by binding of SARM1's own substrate NAD+ in an allosteric location, away from the catalytic sites. This model was validated by mutagenesis of the allosteric site, which led to constitutively active SARM1. We propose that the reduction of cellular NAD+ concentration contributes to the disassembly of SARM1's peripheral ring, which allows formation of active NADase domain dimers, thereby further depleting NAD+ to cause an energetic catastrophe and cell death.

## Introduction

SARM1 (sterile α and HEAT/armadillo motif–containing protein; *Mink et al., 2001*) was first discovered as a negative regulator of TRIF (TIR domain–containing adaptor inducing interferon-β) in TLR (Toll-like receptor) signaling (*Carty et al., 2006*). SARM1 was later shown to promote neuronal death by oxygen and glucose deprivation (*Kim et al., 2007*) and viral infections (*Hou et al., 2013*; *Mukherjee et al., 2013*; *Sundaramoorthy et al., 2020*; *Uccellini et al., 2020*), while also having a protective role against bacterial and fungal infections in *Caenorhabditis elegans* (*Couillault et al., 2004*; *Liberati et al., 2004*). Subsequently, studies have demonstrated that SARM1 is also a key part of a highly conserved axonal death pathway that is activated by nerve injury (*Gerdts et al., 2013*; *Osterloh et al., 2012*). Notably, SARM1 deficiency confers protection against axonal degeneration in several models of neurodegenerative conditions (*Kim et al., 2007*; *Ko et al., 2020*; *Ozaki et al., 2020*; *Uccellini et al., 2020*), making SARM1 a compelling drug target to protect axons in a variety of axonopathies (*Krauss et al., 2020*; *Loring et al., 2020*).

The domain composition of SARM1 includes an N-terminal peptide, an ARM-repeats region, two SAM, and one TIR domain (*Figure 1A*, *Figure 1—figure supplement 1*), which mediate mitochondria targeting (*Panneerselvam et al., 2012*), auto-inhibition (*Chuang and Bargmann, 2005*;

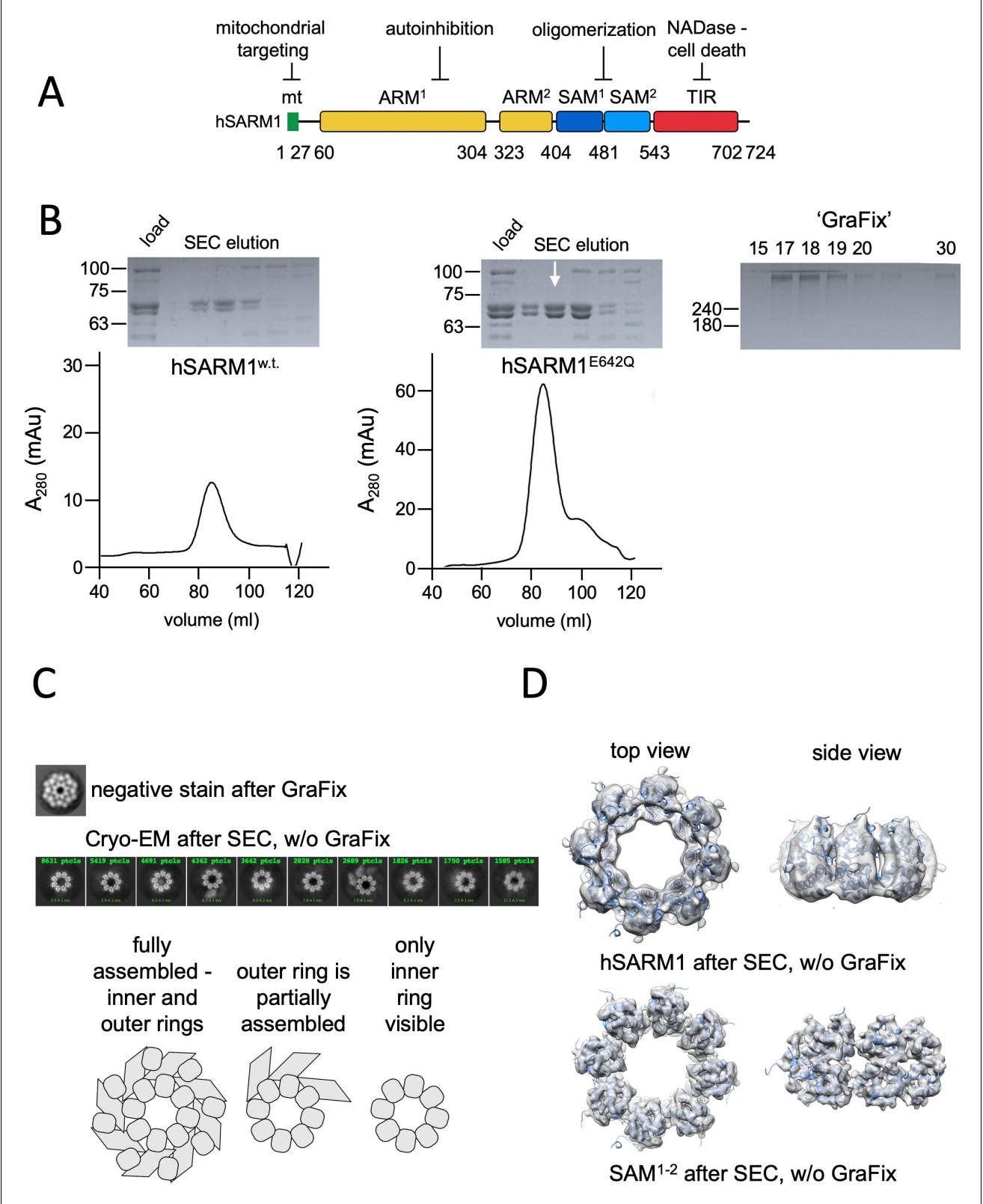

**Figure 1.** Domain organization and cryo-EM analysis of purified hSARM1. (**A**) Color-coded organization and nomenclature of the SARM1 ARM, SAM, and TIR domains. The position of mitochondrial localization signal is presented at the N terminus of the protein. Two constructs are used in this study, both missing the mitochondrial N-terminal sequence. hSARM1$^{E642Q}$ is a NADase attenuated mutant that was used in all the structural and some of the biochemical experiments, while hSARM1$^{w.t.}$ was used in the NADase and cellular experiments. (**B**) hSARM1 protein preparations. Presented are SDS-

*Figure 1 continued on next page*

*Figure 1 continued*

PAGE analyses of size-exclusion chromatography fractions after the initial metal-chelate chromatography step. Note the higher yield of hSARM1$^{E642Q}$ compared to hSARM1$^{w.t.}$, and the doublet SARM1 bands, the result of a partial N-terminal his-tag digestion. The white arrow indicates the fraction used for the subsequent GraFix step shown at the right panel, where the approximate glycerol concentrations of each fraction are indicated. (C) Most-prevalent 2D class averages of hSARM1$^{E642Q}$ protein preparations. A dramatic difference can be seen between the previously conducted negative stain analysis (top panel, as in *Sporny et al., 2019*) and typical cryo-EM (middle panel). While the negative stain average clearly shows inner and peripheral rings (see illustration in the bottom panel), most cryo-EM classes depict only the inner ring, and some a partial outer ring assembly. Note that a gradient fixation (GraFix) protocol was applied before the negative stain but not the cryo-EM measurements. (D) 3D cryo-EM reconstructions of hSARM1$^{E642Q}$ (top panel) and SAM$^{1-2}$ (bottom panel) and docking of the SAM$^{1-2}$ crystal structure (PDB code 6QWV) into the density maps, further demonstrating that only the inner SAM$^{1-2}$ ring is well ordered in cryo-EM of purified hSARM1$^{E642Q}$.

The online version of this article includes the following figure supplement(s) for figure 1:

**Figure supplement 1.** Structure-based sequence alignment of the SARM1 of human, mouse, zebrafish, and the *Caenorhabditis elegans* homolog TIR-1.

*Summers et al., 2016*), oligomerization (*Gerdts et al., 2013*), and NADase activity (*Gerdts et al., 2015*), respectively.

Amino acid substitutions (E642A or E596K) at the TIR domain's active site abolish the NADase activity in vitro and inactivates SARM1 pro-degenerative activity (*Essuman et al., 2017*; *Geisler et al., 2019*; *Horsefield et al., 2019*), thereby linking SARM1's role in axonal degeneration with its NADase activity. The enzymatic activity requires a high local concentration of the TIR domains, as demonstrated by forced dimerization of TIR, which resulted in NAD+ hydrolysis and neuronal cell death (*Gerdts et al., 2015*; *Gerdts et al., 2016*). Also, the deletion of the ARM domain in SARM1, which interacts directly with TIR (*Summers et al., 2016*), renders the 'delARM' construct constitutively active and leads to rapid cell death (*Gerdts et al., 2013*; *Sporny et al., 2019*). How SAM domains cause TIR crowding became clearer in our recent report (*Sporny et al., 2019*), where we showed that both hSARM1 (human SARM1) and the isolated tandem SAM$^{1-2}$ domains form octamers in solution. In this study, we used negative stain electron microscopy analysis of hSARM1 and determined the crystal structure (as did others [*Horsefield et al., 2019*]) of the SAM$^{1-2}$ domains – both of which revealed an octameric ring arrangement.

These findings imply that hSARM1 is kept auto-inhibited by the ARM domain in homeostasis and gains NADase activity upon stress conditions through the following: the infliction of injury (axotomy, *Osterloh et al., 2012*), oxidation (mitochondria depolarization, *Murata et al., 2013*; oxidizing agents, *Summers et al., 2014*), metabolic conditions (depletion of NAD+, *Gilley et al., 2015*), or toxins (chemotherapy drugs, *Geisler et al., 2016*). Whether and how all or some of these insults converge to induce SARM1 activation is still not completely understood. In this regard, little is known about the direct molecular triggers of SARM1 in cells, besides the potential involvement of nicotinamide mononucleotide (NMN) (*Bratkowski et al., 2020*; *Liu et al., 2018a*; *Zhao et al., 2019*) and Ser-548 phosphorylation by c-Jun N-terminal kinase (*Murata et al., 2018*) in promoting the NADase activity of SARM1. Here, we present structural data and complementary biochemical assays to show that SARM1 is kept inactive through a 'substrate inhibition' mechanism, where the high concentration of NAD+ stabilizes the tightly packed, inhibited conformation of the protein. In this way, SARM1 activation is triggered by a decrease in the concentration of a cellular metabolite NAD+, rather than depending on the introduction of an activating factor.

## Results

### Cryo-EM visualization of purified hSARM1

For cryo-EM imaging, we generated a near-intact hSARM1 construct that lacks the N-terminal mitochondrial localization signal ($^{26}$ERL...GPT$^{724}$) and has a point mutation in the NADase catalytic site (E642Q). The resulting construct was expressed in mammalian cell culture and isolated to homogeneity using consecutive metal chelate and size exclusion chromatography steps. For cellular and in vitro activity assays we expressed and isolated hSARM1$^{w.t.}$ ($^{26}$ERL...GPT$^{724}$) without the E642 mutation (*Figure 1B*, *Figure 1—figure supplement 1*), although with lower yields. We first collected cryo-EM images of the purified hSARM1$^{E642Q}$. 2D classification (*Figure 1C*) and 3D reconstruction (*Figure 1D*, *Table 1*) revealed an octamer ring assembly with clearly visible inner ring, which is

**Table 1.** Cryo-EM data acquisition, reconstruction, and model refinement statistics.

| hSARM1 construct and pretreatment | hSARM1 GraFix-ed | hSARM1 +5 mM NAD+ | hSARM1 no treatment | SAM[1–2] |
|---|---|---|---|---|
| Electron microscope | Titan Krios | Titan Krios | F30 Polara | F30 Polara |
| Cryo-EM acquisition and processing | | | | |
| EMDB accession # | 11187 | 11834 | 11190 | 11191 |
| Magnification | 165,000x | 165,000x | 200,000 | 200,000 |
| Voltage (kV) | 300 | 300 | 300 | 300 |
| Total electron exposure (e$^-$/Å$^2$) | 50 | 40 | 80 | 80 |
| Defocus range (μM) | −0.8 to −2.8 | −0.8 to −2.8 | −1.0 to −2.5 | −1.0 to −2.5 |
| Pixel size (Å) | 0.827 | 0.827 | 1.1 | 1.1 |
| Symmetry imposed | C8 | C8 | C8 | C8 |
| Initial particles | 658,575 | 335,526 | 26,496 | 416,980 |
| Final particles | 147,232 | 159,340 | 5410 | 43,868 |
| Resolution (masked FSC = 0.143 Å) | 2.88 | 2.7 | 7.7 | 3.77 |
| Model refinement | | | | |
| PDB ID | 6ZFX | 7ANW | 6ZG0 | 6ZG1 |
| Model resolution (FSC = 0.50/0.143 Å) | 3.6/2.9 | 2.9/2.7 | | |
| Model refinement resolution | 2.88 | 2.70 | | |
| Non-hydrogen atoms | 39,856 | 40,560 | | |
| Residues | 5080 | 5176 | | |
| RMS deviations | | | | |
| Bond length (Å) | 0.008 | 0.009 | | |
| Bond angle (°) | 1.84 | 1.84 | | |
| Ramachandran plot | | | | |
| Favored (%) | 89.05 | 91.68 | | |
| Allowed (%) | 10.95 | 8.32 | | |
| Disallowed (%) | 0 | 0 | | |
| Rotamer outliers (%) | 5.73 | 6.64 | | |
| Validation | | | | |
| MolProbity score | 2.68 | 2.49 | | |
| Clash score | 10.08 | 6.66 | | |

attributed to the tandem SAM domains. Only a minor fraction of the particles (~20%) shows the presence of a (partial) peripheral ring composed by the ARM and TIR domains. Cryo-EM analysis of the isolated SAM[1–2] domains (*Figure 1D*) and docking of the crystal structure of the SAM[1–2] octamer (PDB code 6QWV) into the 3D maps demonstrate that indeed the ARM and TIR domains are largely missing from this reconstruction, implying a disordered peripheral ring in ~80% of the particles. Exploring different buffers, pH and salt conditions, addition of various detergents, as well as variations in cryo-EM grid preparation (e.g. ice thickness) did not affect the visibility of the octamer peripheral ring considerably.

These results are inconsistent with our previous analysis, where we used low resolution negative stain EM visualization and 2D classification of hSARM1[E642Q] that showed fully assembled inner and outer ring structures (*Figure 1C*; *Sporny et al., 2019*). We thought, that a glycerol-gradient fixation step (GraFix; *Kastner et al., 2008*) that involves ultra-centrifugation of the protein sample through a glycerol + glutaraldehyde cross-linker gradients, which was applied before the negative stain – but not before the cryo-EM sample preparations – might be the cause for the difference between the

two imaging conditions. We therefore pursued cryo-EM data collection of GraFix-ed hSARM1[E642Q] after dilution of the glycerol from 18% (which severely diminishes protein contrast in cryo-EM) to 2.5%.

## 2.88 Å resolution structure of a fully assembled compact hSARM1 GraFix-ed octamer

We carried out 2D classification (*Figure 2A*) and 3D reconstruction and refinement (*Figure 2B–D*, *Table 1*) of the GraFix-ed hSARM1[E642Q] to an overall resolution of 2.88 Å (applying eightfold symmetry). The hSARM1[E642Q] octamer is 203 Å in diameter and 80 Å thick (*Figure 2—figure supplement 1A*). The SAM[1–2] domains' inner ring is the best resolved part of the map to which the high-resolution crystal structure (PDB code 6QWV) was fitted with minute adjustments. The TIR domains are the least defined part of the density map. Some parts of the TIR domain reach a local resolution of 6.5 Å (*Figure 2—figure supplement 2A*) and therefore not revealing side chain positions. However, the availability of high-resolution crystal structures of isolated hSARM1 TIR (PDB codes 6O0R, 6O0U; *Horsefield et al., 2019*) allowed their docking into the well-resolved secondary structure elements in the map with very high confidence. The ARM domains show intermediate map quality, with well-resolved secondary structures and bulky side chains. This allowed the building of a de-novo atomic model for the entire ARM, as no high-resolution structure or homology models of SARM1's ARM are available. The entire atomic model is comprised of residues 56–700 (*Figure 1—figure supplement 1*), with an internal break at the linker that connects SAM[2] to the TIR domain. The structural analyses of the SAM domains and the SAM octamer ring assembly, as well as the atomic details of the TIR domain, are described in our previous study and by others (*Horsefield et al., 2019*; *Sporny et al., 2019*). The cryo-EM structure of the ARM domain reveals a closed crescent-shaped region, composed of seven three-helix ARM repeats spanning residues 60–400 (*Figure 2—figure supplement 1B*). The ARM topology is split into two interacting parts, designated ARM[1] (res. 60–303 with five ARM repeats) and ARM[2] (res. 322–400 with two ARM repeats) (*Figure 1—figure supplement 1*). The main ARM[1]–ARM[2] interaction interface is hydrophobic and involves helices α14 and α16 of ARM[1] and helices α1, α2, and α3 of ARM[2]. ARM[1] and ARM[2] are also interacting at the crescent 'horns' through the ARM[1]α2–α3 loop with the loop that connects ARM[1] with ARM[2] (res. 305–320). In the hSARM1 compact octamer, each ARM is directly connected via a linker (res. 400–404) to a SAM[1] domain. Also, each ARM is engaged in several non-covalent interactions with the same-chain SAM and with the clockwise neighboring SAM, when assuming a top view of the structure (*Figure 2B–D*). Although neighboring ARM domains are closely packed, direct interactions between them seem to be limited, engaging a short segment of ARM[1] α9 with the α4–α5 loop of ARM[2] of the neighboring chain. Additional ARM–ARM interactions are indirect, mediated by the TIR domains. Each TIR binds the ARM ring via two sites, designated the 'primary' and 'secondary' TIR docking sites (*Figure 3A*). The 'primary' is larger and engages the TIR helix αA and the EE loop with the ARM[1] α10, α10–α11 loop, and α13. The 'secondary' TIR docking site is smaller and involves the TIR BB loop and helix α7 of the counter-clockwise ARM[1] domain (*Figure 1—figure supplement 1*).

Very recently, *Bratkowski et al., 2020* have reported a 3.3 Å cryo-EM structure of hSARM1[50-724]. Except for a residue registry shift throughout the ARM[1] α1–α4 and a minor one at the ARM[1]–ARM[2] loop, their structure is largely similar to our 2.88 Å GraFix-ed structure. While we have used the Gra-Fix pretreatment to coerce the compact two-ring conformation, it seems that Bratkowski et al. employed a different strategy by selectively using a small subset (5.6%) of the initial picked particles to reconstruct their map. We assume that this was done to cope with the particle heterogeneity of purified SARM1. An obstacle we have encountered as well, where we observed that ~80% of the particles have a disordered outer ring.

## The compact conformation of hSARM1 is inhibited for NADase activity

The GraFix-ed cryo-EM structure revealed a domain organization where the catalytic TIR subunits are separated from each other by docking onto the ARM peripheral ring. This assembly is dependent on the coupling of each TIR with two neighboring ARM domains (*Figure 3A*). Since the NADase activity requires close proximity of several TIR domains (*Bratkowski et al., 2020*; *Gerdts et al., 2015*; *Gerdts et al., 2016*), and possibly homo-dimerization (*Horsefield et al., 2019*), we considered that the cryo-EM structure represents an inhibited conformation of SARM1, in which

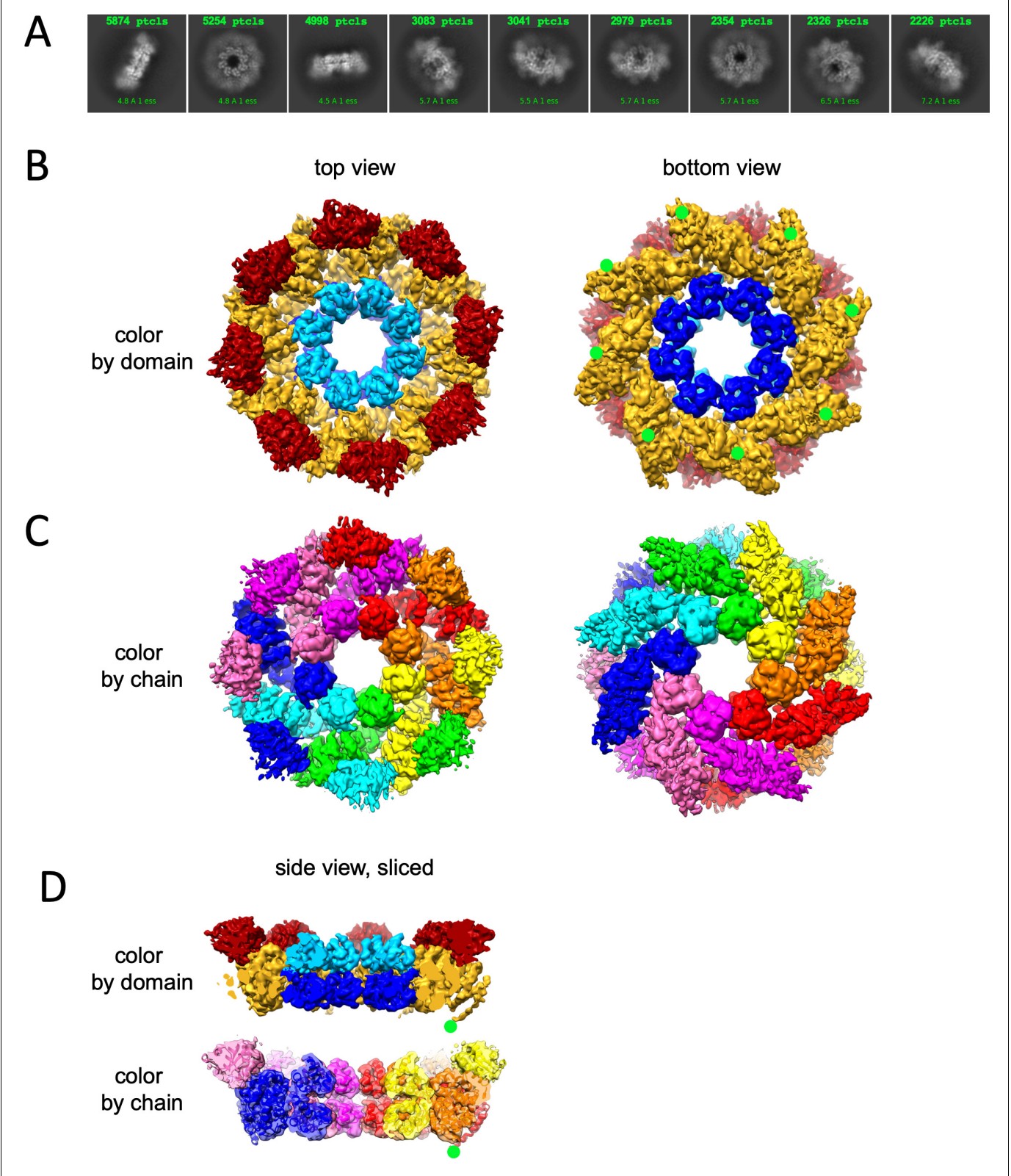

**Figure 2.** Cryo-EM structure of GraFix-ed hSARM1$^{E642Q}$. (A) Selected representation of 2D class averages used for the 3D reconstruction. The number of particles that were included in each average is indicated at the top of each class. (B–D) Cryo-EM density map color coded as in *Figure 1A* and by chain. 'Top view' refers to the aspect of the molecule showing the TIR (red) and SAM$^2$ (cyan) domains closest to the viewer, while in the 'bottom view' the SAM$^1$ (blue) domains and the illustrated mitochondrial N-terminal localization tag (green dot – was not included in the expression construct) are the

*Figure 2 continued on next page*

*Figure 2 continued*

closest. (D) Side view representation of the structure, sliced at the frontal plane of the aspect presented in (B) and (C). Note that while the GraFix-ed map has an overall lower resolution, it has a heterogenous distribution with distinctive differences between the SAM, ARM, and TIR regions. On the contrary, in the NAD+ supplemented map, resolution values are much more homogenous.

The online version of this article includes the following figure supplement(s) for figure 2:

**Figure supplement 1.** Structure and domain organisation of hSARM1.

**Figure supplement 2.** Resolution, angular distribution, and B-factor estimations of the Cryo-EM maps of GraFix-ed (upper panel) and NAD+ supplemented (bottom panel) hSARM1.

the TIR domains cannot form compact dimers and catabolize NAD+. To test this hypothesis, we aimed to weaken TIR docking to allow their nearing and subsequent NAD+ catalysis. To this end we designed amino acid substitutions at the ARM's primary TIR docking site without compromising the protein's structural integrity – particularly that of the TIR domain (*Figure 3A*). Two pairs of mutations, RR216-7EE of ARM[1] helix α10 and FP255-6RR of ARM[1] helix α13 (*Figure 3C*, *Figure 1—figure supplement 1*), were introduced, as well as the double mutant RR216-7EE/FP255-6RR. These constructs were transiently expressed in HEK-293T cells. The effect of these constructs on NAD+ levels and cell viability were monitored using a previously established resazurin fluorescence assay (*Essuman et al., 2018*; *Gerdts et al., 2013*; *Sporny et al., 2019*). The results (*Figure 3C*) show a rapid decrease in cellular NAD+ levels and 50% cell death within 24 hr after transfection of the FP255-6RR and double mutant. This toxicity level is similar to that of a hSARM1 construct missing the entire ARM domain 'delARM' (res. 409–724) (*Figure 3C*), proven to be toxic in neurons and HEK293 cells (*Gerdts et al., 2013*; *Sporny et al., 2019*). This toxic effect was attributed to the removal of auto-inhibitory constraints imposed by the ARM domain. The RR216-7EE mutation has a weaker effect, probably due to the position of these amino acid residues at the margin of the TIR-ARM interface.

In conclusion, we found that hSARM1 inhibition requires ARM-TIR interaction through the 'primary TIR docking site'. This conclusion is supported by other recently published reports (*Bratkowski et al., 2020*; *Jiang et al., 2020*). We further calculated the surface conservation scores of SARM1 orthologs. The scores were color coded and plotted on the molecular surface of hSARM1 using the Consurf server (*Figure 3B*; *Ashkenazy et al., 2016*). The surface-exposed residues reveal a high level of conservation on both the ARM and TIR domain–binding interface. This indicates the biological importance of the interface and a possible conservation of its function in auto-inhibition among species. It was previously suggested that auto-inhibition of SARM1 is maintained by keeping SARM1 as a monomer, and that upon activation SARM1 assembles into a multimer (*Figley and DiAntonio, 2020*), very much like other apoptotic complexes. Our results show otherwise and explain how hSARM1 avoids premature activation even as a pre-formed octamer, while kept poised for rapid activation and response.

## Isolated hSARM1 is NADase active in vitro and inhibited by glycerol

As it became clear that the compact two-ring structure is inhibited for NADase activity, we considered whether the purified hSARM1, that was not subjected to GraFix and predominantly presents just the inner SAM ring in cryo-EM 2D averaging and 3D reconstruction (*Figure 1C and D*), is active in vitro. Using a resazurin fluorescence assay modified for an in vitro application, we measured the rate of NAD+ consumption by hSARM1[w.t.] in a series of NAD+ concentrations and determined a $K_m$ of 28 ± 4 μM, with $V_{max}$ of 9 ± 0.3 μM/min and $K_{cat}$ of 46.49 1/min (*Figure 3D*). These kinetic parameters are remarkably similar to a published $K_m$ of 24 μM (*Essuman et al., 2017*) – especially when considering that the quoted study used an isolated TIR domain fused to artificial dimerizing and aggregating agents, while we used the near-intact protein.

It was previously discovered that NMN levels rise after axonal injury, and that exogenous increase of NMN induces degeneration (*Di Stefano et al., 2015*). Moreover, a membrane-permeable NMN analogue activates SARM1 in cultured cells, leading to their death (*Liu et al., 2018a*; *Zhao et al., 2019*). We therefore measured the effect of NMN supplement on the NADase activity of purified hSARM1 and found a moderate 30% increase in activity with 1 mM NMN, but none at a lower concentration of 200 μM (*Figure 3E*). This demonstrates that the purified hSARM1 is mostly NADase

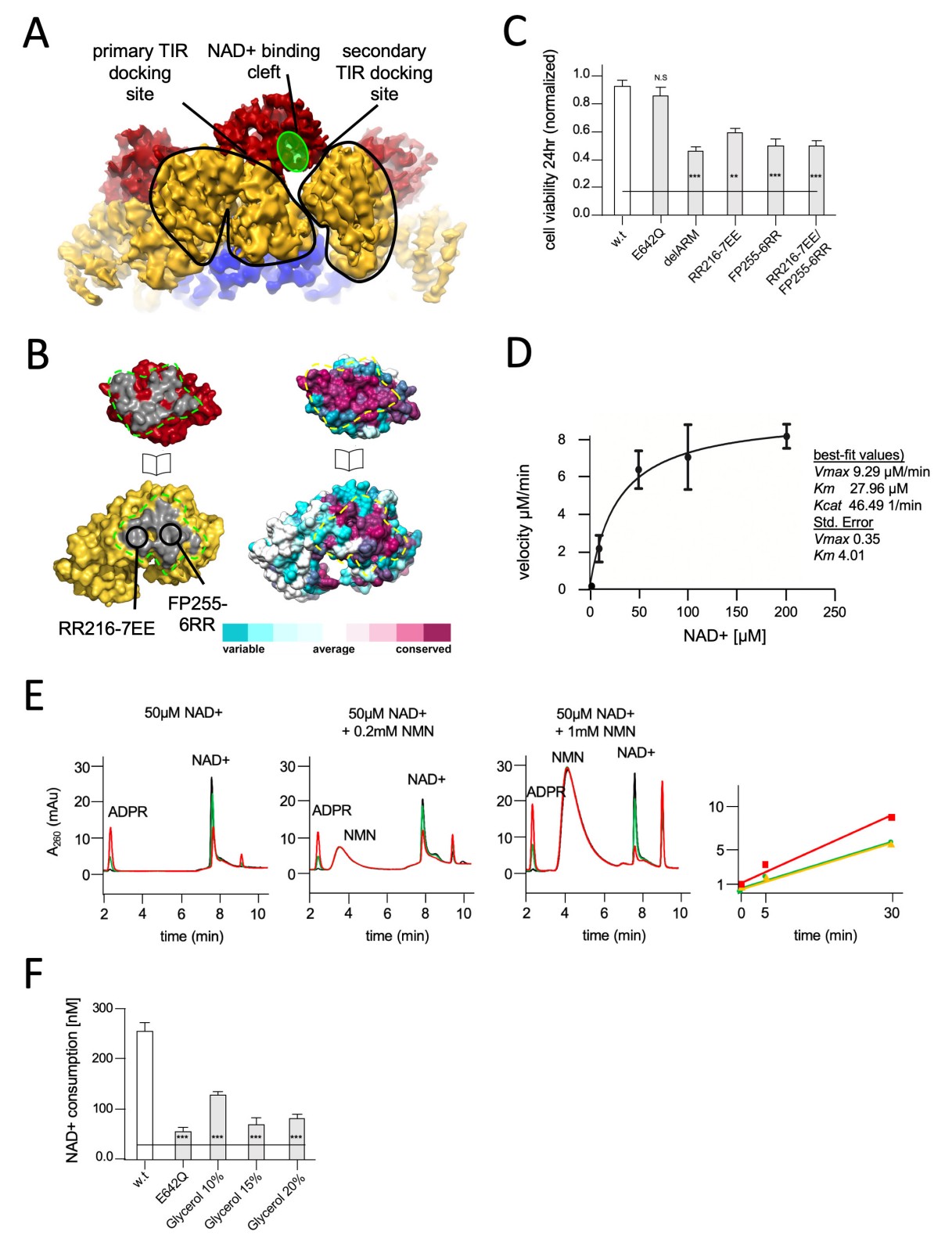

**Figure 3.** Structural basis for hSARM1 auto-inhibition. (**A**) Close-up of a tilted side view of the GraFix-ed hSARM1$^{E642Q}$ map (colored as in *Figure 2A*). Two neighboring ARM domains (yellow) are outlined by a black line and the NAD+ binding cleft of a TIR domain that is bound to the two ARMs is highlighted in green. The interfaces formed between the TIR and ARMs are designated as the 'primary TIR docking site' and the 'secondary TIR docking site'. (**B**) Two 'open-book' representations (in the same orientation) of the 'primary TIR docking site'. Left – the TIR and ARM domains are

*Figure 3 continued on next page*

*Figure 3 continued*

colored as in (**A**), and the interface surfaces in gray are encircled by green dashed line. Site-directed mutagenesis sites are indicated. Right –– amino acid conservation at the 'primary TIR docking site'. Cyan through maroon are used to indicate amino acids, from variable to conserved, demonstrating an overall high level of conservation in this interface. (**C**) Toxicity of the hSARM1 construct and mutants in HEK293T cells. The cells were transfected with hSARM1 expression vectors, as indicated. Cell viability was measured and quantified 24 hr post-transfection using the fluorescent resazurin assay. While cell viability is virtually unaffected after 24 hr by ectopic expression of hSARM1$^{w.t.}$ and hSARM1$^{E642Q}$, deletion of the inhibiting ARM domain (which results in the SAM$^{1-2}$–TIR construct) induces massive cell death. Mutations at the 'primary TIR docking site' of the ARM domain also induce cell death, similar to the 'delARM' construct (three biological repeats, Student's t-test; ***p<0.001; *p<0.05; n.s: no significance). (**D**) Kinetic measurement of purified hSARM1 NADase activity. $K_m$ and $V_{max}$ were determined by fitting the data to the Michaelis–Menten equation and are presented as mean ± SEM for three independent measurements. (**E**) HPLC analysis of time-dependent NAD+ consumption (50 µM) by hSARM1, and further activation by nicotinamide mononucleotide (NMN) (0.2 and 1 mM). Time points 0 (black), 5 (green), and 30 (red) min. Right graph shows the rate of adenosine diphosphate ribose (ADPR) product generation: no NMN (green), 0.2 mM NMN (orange), and 1 mM NMN (red). While 0.2 mM NMN has no visible effect, 1 mM NMN increases hSARM1 activity by ~30%. (**F**) Glycerol inhibition of hSARM1 NADase in vitro activity was measured 20 min after adding 0.5 µM NAD+ to the reaction mixture. In the same way, hSARM1$^{E642Q}$ activity was measured and compared to that of hSARM1$^{w.t.}$, showing attenuated NADase activity of the former.

active, even without NMN supplement, supporting the idea of a predominantly open conformation as seen in cryo-EM (*Figure 1C and D*).

Next, we examined what confines the GraFix-ed hSARM1 into the compact inhibited conformation (*Figure 2*) and measured the in vitro NADase activity in the presence of glycerol (*Figure 3F*). We found that glycerol reduces hSARM1 NADase activity in a concentration-dependent manner, reaching 72% inhibition at 15% glycerol. In the GraFix preparation, we extracted hSARM1 from the gradient after it migrated to approximately 18% glycerol concentration. This is the position we used to image SARM1 in its inhibited compact conformation. It seems likely that this conformation is maintained after glycerol is removed, due to glutaraldehyde crosslinking which preserves the compact structure.

## NAD+ substrate inhibition of hSARM1

Our results show that hSARM1 NADase activity is suppressed in cell culture, but much less so in vitro after being isolated. This prompted us to hypothesize that in the course of purification from the cytosolic fraction, hSARM1 loses a low-affinity cellular factor, responsible for inhibiting it in the cellular environment. SARM1 was previously shown to be activated in cell culture in response to metabolic, toxic, and oxidative stressors. We reasoned that the hypothesized inhibitory factor is a small molecule that is depleted under cellular stress conditions, and thus the inhibition of hSARM1 is released. To follow through on this hypothesis, we tested the impact of several small molecules, which are associated with the cell's energetic state, with two in vitro parameters: hSARM1 NADase activity and structural conformation (visualized by cryo-EM). For instance, we already established that glycerol meets these two criteria, as it imposes hSARM1 compact conformation (*Figure 2*) and reduces NADase activity (*Figure 3F*). Curiously, glycerol was found to occupy the active site of the hSARM1 TIR domain in a crystal structure (PDB code 6O0R [*Horsefield et al., 2019*]), directly linking structural data with enzymatic inhibition. However, the glycerol concentrations in which these in vitro experiments were conducted were very high (15–18% for the NADase activity assay and cryo-EM, and 25–30% in the X-ray crystal structure), that is, 2–4 M, which is considerably higher than the estimated sub 1 mM concentrations of intracellular physiological glycerol (*Li and Lin, 1983*). Next, we considered ATP as a fit candidate to inhibit hSARM1, because ATP is the main energetic compound, with cellular concentrations between 1 and 10 mM, which is depleted prior to cell death and axonal degeneration. Indeed, we found that ATP inhibits the NADase activity of hSARM1 in a dose-dependent manner (*Figure 4A*). However, it did not affect hSARM1 conformation as observed by 2D classification (*Figure 4B*). Therefore, it is possible that ATP inhibits hSARM1 through the TIR domain's active site.

It was reported that NAD+ levels drop in response to axon injury (*Coleman and Höke, 2020*). Also, the ablation of the cytosolic NAD+ synthesizing enzymes NMNAT1 and 2 decreases cytoplasmic NAD+ and induces SARM1 activation (*Gilley et al., 2015*; *Sasaki et al., 2016*). Therefore, we postulated that it is the high physiological concentration of NAD+ itself that may inhibit hSARM1 through 'substrate inhibition' – a general mechanism regulating the activity of many enzymes

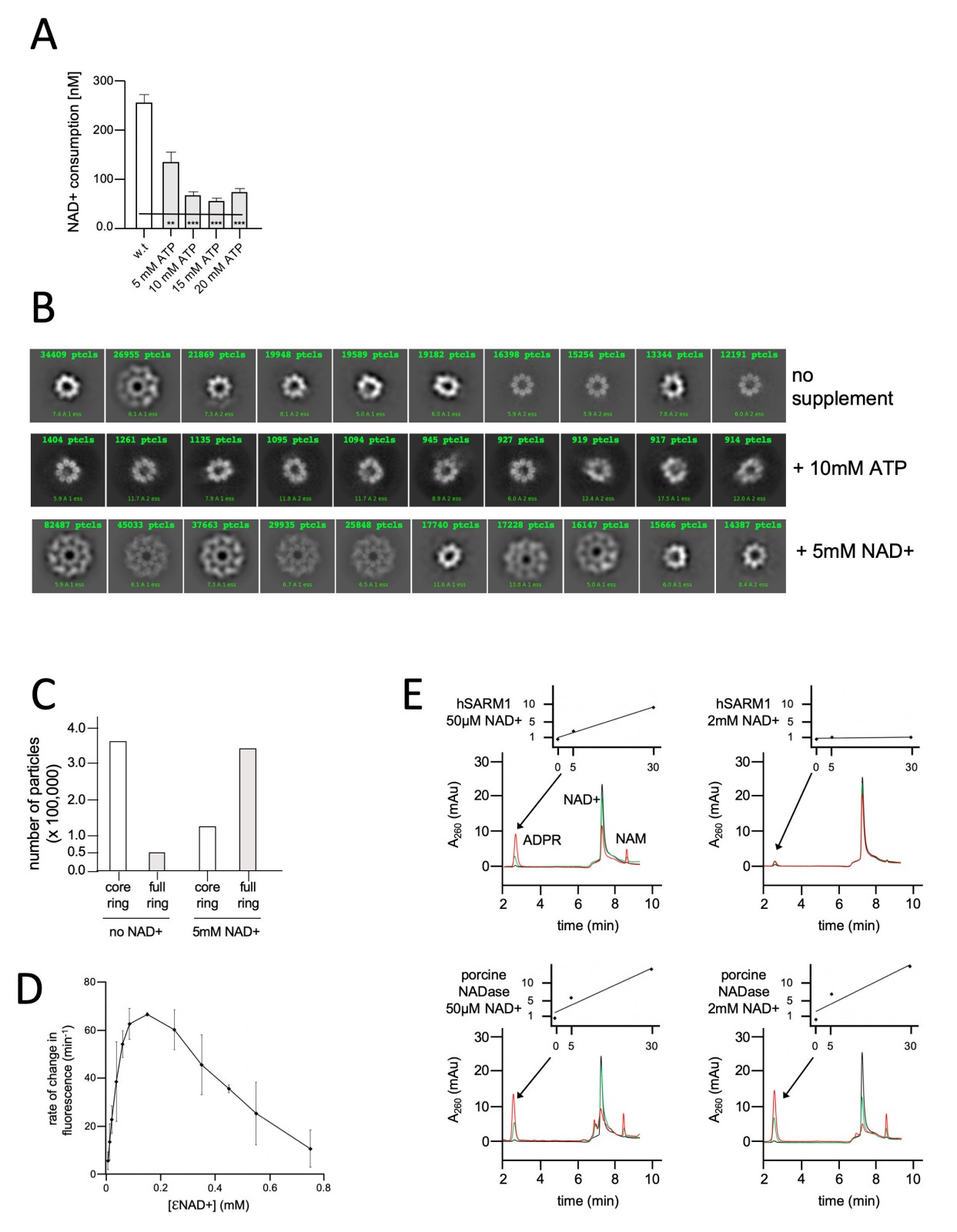

**Figure 4.** NAD+ induces structural and enzymatic inhibition of hSARM1. (**A**) Inhibition of hSARM1 NADase activity by ATP was demonstrated and analyzed as in *Figure 3F*. (**B**) The structural effects of NAD+ and ATP were observed by cryo-EM based on appearance in 2D classification. Presented are the 10 most populated classes (those with the largest number of particles – numbers in green, first class is on the left) out of 50–100 from each data collection after the first round of particle picking and classification. By this analysis, the percentage of particles that present full, two-ring structure is

*Figure 4 continued on next page*

*Figure 4 continued*

13% (no NAD+); 74% (5mM NAD+); and 4% (10mM ATP). (**C**) Total number of particles with full ring assembly vs. those where only the inner ring is visible. Conditions of sample preparation, freezing, collection, and processing were identical, except for the NAD+ supplement in one of the samples. (**D**) Rate of change in nucleotide fluorescence under steady-state conditions of eNAD hydrolysis by hSARM1. Reactions were initiated by mixing 400 nM enzyme with different concentrations of 1:10 eNAD+:NAD+ (mol/mol) mixtures. Three repeats, standard deviation error bars. (**E**) HPLC analysis of time-dependent NAD+ consumption by hSARM1 and porcine brain NADase control in 50 µM and 2 mM. Time points 0 (black), 5 (green), and 30 (red) min. Inset graph shows the rate of ADPR product generation. While the rate of NAD+ hydrolysis by porcine NADase is maintained through 50 µM and 2 mM NAD+, hSARM1 is tightly inhibited by 2 mM NAD+.

(*Reed et al., 2010*). Cryo-EM images show that adding 5 mM NAD+ to hSARM1 has a dramatic effect: >80% of the particles show the two-ring, compact conformation – in contrast to the <20% found in the absence of NAD+ (*Figure 4B and C*). To measure NAD+ substrate inhibition of hSARM1, we used a fluorescent assay with a wide concentration range of the NAD+ analog, etheno-NAD (eNAD) (mixed 1:10 mol/mol with regular NAD+). eNAD fluoresces upon hydrolysis (ex. λ = 330 nm, em. λ = 405 nm). The results showed a bell-shaped curve (*Figure 4D*) with the highest rate of hydrolysis at 100 µM NAD+ and a steady decrease in rate thereafter (at the higher concentrations). In addition, we used a direct reverse-phase HPLC monitoring of NAD+ consumption by hSARM1 compared with commercially available porcine brain NADase. While the rate of hydrolysis of porcine NADase was maintained between 50 µM and 2 mM NAD+, hSARM1 was almost completely inhibited at 2 mM NAD+ (*Figure 4E*).

## 2.7 Å resolution structure of NAD+ induced hSARM1 compact octamer

Following evidence for NAD+ substrate inhibition of hSARM1 by cryo-EM 2D classification (*Figure 4B and C*) and enzymatic (*Figure 4D and E*) assays, we pursued a 3D structure determination of hSARM1 complexed with NAD+ at inhibiting concentrations. hSARM1, supplemented by 5 mM NAD+, was imaged, reconstructed, and refined to 2.7 Å resolution (*Figure 5A*), resulting in an excellent map (*Figure 5* and *Figure 5—figure supplement 1*). The structure was largely identical to the GraFix-ed structure, further substantiating the validity of the latter (*Figure 5B*). Few structural differences can be observed between the NAD+ supplemented density map and that of the GraFixed hSARM1. The first difference is a 5 Å shift in the position of the distal part of the TIR domain, and the second is a rearrangement of the secondary structure at the region of the 'crescent horns', where the tips of ARM[1] and ARM[2] touch (*Figure 5C*, *Figure 5—figure supplements 1* and *2A and B*). In the same region, designated hereafter 'ARM allosteric site', an extra-density in the NAD+ supplemented map reveals the binding site of one NAD+ molecule, providing clear atomic details (*Figure 5C*, right panel). To probe into the function of the ARM allosteric site, we introduced point mutations, targeting three structural elements surrounding the NAD+ density. These elements were ARM[1] α2, ARM[1] α5–α6 loop, and the ARM[1]–ARM[2] loop (*Figure 5C and D*, *Figure 5—figure supplement 2*). As a control, we introduced two other mutations in sites that we do not consider to be involved in hSARM1 inhibition or activation. We hypothesized that NAD+ binding at the ARM allosteric site stabilizes the ARM conformation by interacting with both ARM[1] and ARM[2], thereby promoting the hSARM1 compact auto-inhibited structure. Therefore, mutations in the allosteric site, which interfere with NAD+ binding, would diminish auto-inhibition and allow hSARM1 activity in cell culture. Indeed, we found that two separate mutations in the ARM[1] α5–α6 loop (L152A and R157E) and one in the opposite ARM[1]–ARM[2] loop (R322E) had a dramatic effect on hSARM1 activity, promoting cell death levels comparable to those induced by the 'delARM' 'constitutively active' construct, which is missing the entire ARM domain (*Figure 5D*). Two mutations (D314A and Q320A), which are also located at the ARM[1]–ARM[2] loop but their side chains positioned away from the NAD+ molecule, did not induce hSARM1 activation. As expected, the control mutations E94R and K363A did not affect hSARM1 activity either. Surprisingly, mutating the bulky W103, which stacks with the NAD+ nicotinamide ring, into an alanine had only a small effect. Nevertheless, W103D mutation did have strong activating effect, leading to the notion that NAD+ contacts with the W103 side chain may be dispensable for the NAD+ inhibitory effect, but interference (as by the aspartate side chain) cannot be tolerated.

Interestingly, we did not find a density indicating the presence of an NAD+ molecule at the TIR domain active site, although this site is not occluded by the ARM domain (*Figure 5—figure*

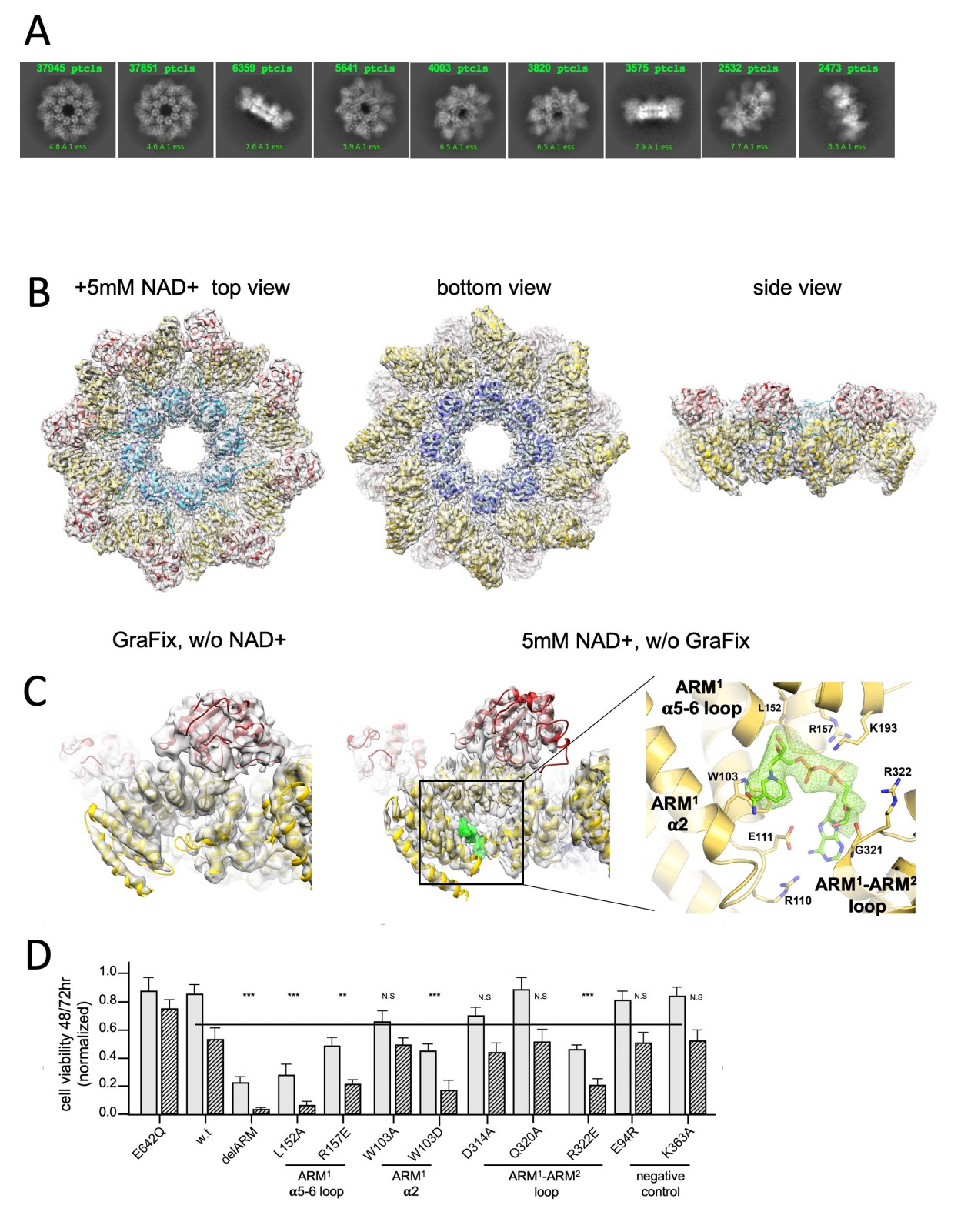

**Figure 5.** 3D structure reveals an inhibitory ARM allosteric NAD+ binding site. (**A**) Selected representation of 2D class averages used for the 3D reconstruction of hSARM1[E642Q] supplemented with 5 mM of NAD+. The number of particles that were included in each class average is indicated. (**B**) Color coded (as in *Figure 1A*) protein model docked in a transparent 2.7 Å cryo-EM density map (gray). (**C**) When compared to the GraFix-ed map, without NAD+ supplement (left), an extra density appears at the 'ARM horns' region in the NAD+ supplemented map (middle, right). The extra density

*Figure 5 continued on next page*

*Figure 5 continued*

is rendered in green and an NAD+ molecule is fitted to it. The NAD+ is surrounded by three structural elements, as indicated on the right panel. The NAD+ directly interacts with the surrounding residues: the nicotinamide moiety is stacked onto the W103 side chain rings; the following ribose with L152 and H190; R110 forms salt bridge with phosphate alpha (proximal to the nicotinamide), while the beta phosphate (distal to the nicotinamide) forms salt bridges with R110 and R157. The map density at the distal ribose and adenosine moieties is less sharp, but clearly involves interactions with the ARM[2] R322, G323, and D326. In this way, activation by nicotinamide mononucleotide, that lacks the distal phosphate, ribose, and adenosine, can be explained by binding to ARM[1] while preventing the bridging interactions with ARM[2]. (D) Toxicity of the hSARM1 construct and mutants in HEK293F cells. The cells were transfected with hSARM1 expression vectors, as indicated. Viable cells were counted 48 (bars in gray) and 72 (bars in black stripes) hr post-transfection. Moderate reduction in cell viability due to ectopic expression of hSARM1[w.t.] becomes apparent 72 hr after transfection, when compared with the NADase attenuated hSARM1[E642Q], while the 'delARM' construct marks a constitutive activity that brings about almost complete cell death after 3 days. Mutations at the ARM[1] α5–α6 and ARM[1]–ARM[2] loops induce cell death at a similar level as the 'delARM' construct, while the control mutations and W103A did not show increased activity (three biological repeats, Student's t-test; ***p<0.001; *p<0.05; n.s: no significance).

The online version of this article includes the following figure supplement(s) for figure 5:

**Figure supplement 1.** Cartoon model and density map (in transparent gray) of the NAD+ supplemented hSARM1.
**Figure supplement 2.** Zooming-in on the ARM allosteric NAD+ binding site.

---

*supplement 1B*). Possibly, the BB loop, which is interacting with a neighboring ARM domain (*Figure 5—figure supplement 1B*), assumes a conformation that prevents NAD+ entry into the binding cleft.

Another conspicuous distinction between the two maps is the difference in local resolution of the domains. The difference between the highest and lowest resolution domains (SAM > ARM > TIR) is more obvious in the GraFix-ed structure than the NAD+ supplemented structure (*Figure 2—figure supplement 2*). This implies that there is more flexibility in the GraFix-ed structure. A likely reason for the difference in map homogeneity is that while the compact arrangement in the GraFix-ed structure is artificially imposed by high glycerol concentration, the NAD+ supplement seems to induce a more natural compact folding.

## Discussion

Our octamer ring structure of a near-intact hSARM1 reveals an inhibited conformation, in which the catalytic TIR domains are kept apart from each other, unable to form close homodimers, which are required for their NADase activity. This inhibited conformation readily disassembles and gains most of its potential activity during protein purification, substantiating our hypothesis that a low-affinity cellular factor inhibits hSARM1 is lost in purification. We have tested a few candidate molecules and found that NAD+ induces a dramatic conformational shift in purified hSARM1, from a disassembled outer ring to a compact two-ring structure, through binding to a distal allosteric site from the TIR catalytic domain. Point mutations in this distal site promoted hSARM1 activity in cultured cells and demonstrates their key allosteric role for inhibition of the NADase activity. We also found that hSARM1 is inhibited in vitro by NAD+ for NADase activity, demonstrating a 'substrate inhibition mechanism', as was also reported by another recent study (*Jiang et al., 2020*).

Following these results, we propose a model for hSARM1 inhibition in homeostasis and activation under stress (*Figure 6*). In this model, we suggest that hSARM1 is kept inhibited by NAD+ through its allosteric site, located at the ARM domain 'horns' junction that induces the compact inhibited conformation. This state persists as long as NAD+ remains at normal cellular levels (these are controversial and range between 0.1 mM [*Cambronne et al., 2016*] and 0.4–0.8 mM [*Hara et al., 2019*; *Houtkooper et al., 2010*; *Liu et al., 2018b*]). However, upon a drop in the cellular NAD+ concentration below a critical threshold, such as in response to stress, NAD+ dissociates from the allosteric inhibitory site. This triggers the disassembly of the compact conformation and the dimerization of TIR domains, which enables TIR's NADase activity and NAD+ hydrolysis. The consequence is a rapid decrease in NAD+ cellular levels, leading to energetic catastrophe and cell death.

This model seems to be at odds with another suggested mechanism (*Bratkowski et al., 2020*; *Zhao et al., 2019*) in which, following axon injury, an increase in NMN concentration triggers SARM1 activation. Regardless of the question whether axon injury actually leads to lasting NMN levels elevation (*Di Stefano et al., 2015*) or not (*Sasaki et al., 2016*), we believe that the two models can be easily reconciled, at least in the limited scope of the in vitro enzymatic mechanism. It is likely

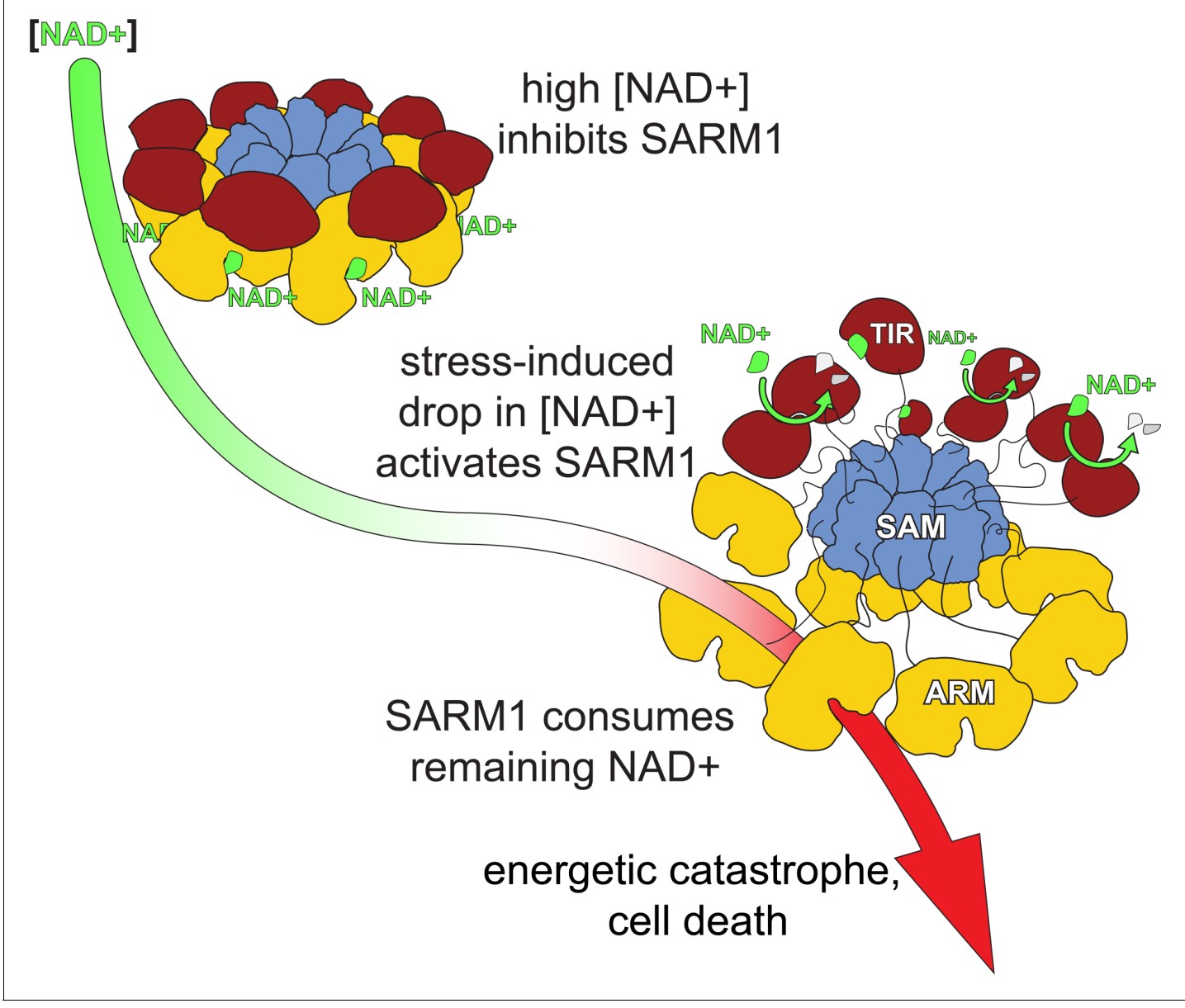

**Figure 6.** A model for hSARM1 inhibition and activation. In homeostasis, the cellular NAD+ concentration is high enough and binds to an allosteric site that drives hSARM1 compact conformation. In this conformation, the catalytic TIR domains (red) are docked on ARM domains (yellow) apart from each other and unable to form close dimers required for NAD+ catalysis. When cellular NAD+ levels drop as a result of reduced NAD+ synthesis (e.g. inhibition of NMNAT1/2) or increased NAD+ consumption, the inhibiting NAD+ molecules fall off hSARM1, leading to the disintegration of the ARM-TIR outer ring assembly. Still held by the constitutively assembled SAM inner ring, the now-released TIR domains are at a high local concentration that facilitates their dimerization and ensued NADase activity. When released from allosteric inhibition, hSARM1 is only subjected to competitive inhibition such as by its products ADPR and NAM, which are not found in high enough concentrations to block its activity entirely. This leads to an almost complete consumption of the NAD+ cellular pool and to an energetic catastrophe from which there may be no return.

that NMN, a moiety of NAD+, will form interactions with most of the ARM[1] residues as NAD+. By that, NMN will compete with NAD+ and prevent the formation of ARM[1]–ARM[2] NAD+ bridge, the latter stabilizing the inactive conformation of SARM1. Such NAD+ displacement would result with SARM1 NADase activation. Clearly, this will require high NMN concentrations to effectively compete-off the ARM[1]–ARM[2] bound NAD+. Indeed, we show (*Figure 3E*) that only 1 mM NMN (but not 0.2 mM) induces a moderate increase in SARM1 NADase activity at 50 µM NAD+.

This finding casts some doubt over the prospects for activation by NMN to be naturally occurring in vivo, as it consistently appears that NMN concentrations are not higher than 5 µM, and in any case 10–200 times lower than that of NAD+ (*Di Stefano et al., 2015*; *Liu et al., 2018b*; *Sasaki et al., 2016*). Therefore, while NMN could activate SARM1 in vitro in cultured cells engineered to over produce NMN, or by exogenous supplement of NMN or NMN analogs, it is less likely to occur in vivo, suggesting that the release of substrate inhibition due to the drop in NAD+ levels is the key event in SARM1 activation.

# Materials and methods

## Key resources table

| Reagent type (species) or resource | Designation | Source or reference | Identifiers | Additional information |
|---|---|---|---|---|
| Gene (*Homo sapiens*) | SARM1 | Imagene | uniprot Q6SZW1 | |
| Cell line (*Homo sapiens*) | HEK293F | Thermo Fisher Scientific | Cat#R79007 | |
| Cell line (*Homo sapiens*) | HEK293T | ATCC | CRL-11268 | |
| Transfected construct (mammalian vector) | pEGFP-N1 | Clontech | | modified |
| Chemical compound, drug | Polyethyleneimine 40 kDa | Polysciences | PEI-MAX | |
| Chemical compound, drug | Resazurin sodium salt | SIGMA | R7017 | |
| Chemical compound, drug | NAD+ | SIGMA | NAD100-RO | |
| Chemical compound, drug | FMN | SIGMA | F8399 | |
| Chemical compound, drug | ADH | SIGMA | A3263 | |
| Chemical compound, drug | Diaphorase | SIGMA | D5540 | |
| Chemical compound, drug | BSA | ORNAT | B9001S | |
| Chemical compound, drug | eNAD | SIGMA | N2630 | |
| Software, algorithm | MotionCorr2 | *Li et al., 2013* | | |
| Software, algorithm | Gctf | *Zhang, 2016* | | |
| Software, algorithm | Cryosparc V2 | *Punjani et al., 2017* | | |
| Software, algorithm | SCIPION wrapper | *Martínez et al., 2020* | | |
| Software, algorithm | Warp | *Tegunov and Cramer, 2019* | | |
| Software, algorithm | CCP4 | *Winn et al., 2011* | | |
| Software, algorithm | GESAMT | *Krissinel, 2012* | | |
| Software, algorithm | MOLREP | *Vagin and Teplyakov, 2010* | | |
| Software, algorithm | Coot | *Emsley et al., 2010* | | |
| Software, algorithm | REFMAC5 | *Murshudov et al., 2011* | | |
| Software, algorithm | JLIGAND | *Lebedev et al., 2012* | | |

## cDNA generation and subcloning

Cloning of all the constructs was made by PCR amplification from the complete cDNA clone (Imagene) of hSARM1 (uniprot: Q6SZW1). For expression in mammalian cell culture, the near-intact

hSARM1$^{w.t.}$ ($^{26}$ERL...GPT$^{724}$) and the mutants hSARM1$^{E642Q}$, hSARM1$^{RR216-7EE}$, hSARM$^{FP255-6RR}$, hSARM1$^{RR216-7EE/FP255-6RR}$, and delARM ($^{387}$SAL...GPT$^{724}$) constructs were ligated into a modified pEGFP-N1 mammalian expression plasmid which is missing the C-terminus GFP fusion protein, and includes N-terminal 6*HIS-Tag followed by a TEV digestion sequence. Assembly PCR mutagenesis (based on https://openwetware.org/wiki/Assembly_pcr) was used to introduce all the point mutations.

## Protein expression and purification

For protein purification, SARM1$^{w.t.}$ and SARM1$^{E642Q}$ were expressed in HEK293F suspension cell culture, grown in FreeStyle 293 medium (GIBCO), at 37°C and in 8% CO$_2$. Transfection was carried out using preheated (70°C) 40 kDa polyethyleneimine (PEI-MAX) (Polysciences) at 1 mg of plasmid DNA per 1 l of culture once cell density has reached $1 \times 10^6$ cells/ml. Cells were harvested 4 (in the case of SARM1$^{w.t.}$) and 5 (in the case of SARM1 $^{E642Q}$) days after transfection by centrifugation (10 min, 1500 × g, 4°C), resuspended with buffer A (50 mM phosphate buffer pH 8, 400 mM NaCl, 5% glycerol, 1 mM DTT, 0.5 mM EDTA, protease inhibitor cocktail from Roche) and lysed using a microfluidizer followed by two cycles of centrifugation (12,000 × g 20 min). Supernatant was then filtered with 45 µm filter and loaded onto a pre-equilibrated Ni-chelate column. The column was washed with buffer A supplemented with 25 mM Imidazole until a stable baseline was achieved. Elution was then carried out in one step of 175 mM Imidazole, after which protein-containing fractions were pooled and loaded onto pre-equilibrated Superdex 200 HiLoad 16/60 (GE Healthcare) for size exclusion chromatography and elution was performed with 25 mM phosphate buffer pH 8.5, 120 mM NaCl, 2.5% glycerol, and 1 mM DTT. In SDS-PAGE Coomassie staining, SARM1 appears as a doublet band (*Figure 1B*). This is probably a result of N-terminal His-tag truncation in some of the SARM1 protein molecules, due to our observation that only the upper band appears in anti-His WB. Protein-containing fractions were pooled and concentrated using a spin concentrator to 1.5 mg/ml. The concentrated proteins were either split into aliquots, flash-frozen in liquid N$_2$, and stored at −80°C for later cryo-EM visualization and enzymatic assays, or immediately subjected to a 'GraFix' (*Kastner et al., 2008*) procedure as follows. Ultracentrifugation was carried out using a SW41Ti rotor at 35,000 rpm for 16 hr at 4°C in a 10–30% glycerol gradient (prepared with gradient master-ip BioComp Instruments, Fredericton, Canada), with a parallel 0–0.2% glutaraldehyde gradient, and with buffer composition of 25 mM phosphate buffer pH 8.5, 120 mM NaCl, and 1 mM DTT. The protein solution volume that was applied to GraFix was 0.4 ml. After ultracentrifugation, the 12 ml gradient tube content was carefully fractionated into 0.75 ml fractions, supplemented with 10 mM aspartate pH 8 to quench crosslinking, using a regular pipette. Most of the cross-linked protein was found at the fractions around 18% glycerol with minor amount at the bottom of the tube (see *Figure 1B*). Analysis was made by SDS-PAGE and the dominant two to three fractions were pooled and diluted by 25 mM phosphate buffer pH 8.5, 120 mM NaCl, and 1 mM DTT, so to reach a final glycerol concentration of 2.5%. The diluted sample was then concentrated using a 100 KDa cutoff Centricon spin concentrator to reach 1 mg/ml protein concentration.

## Cryo-EM grids preparation

Cryo-EM grids were prepared by applying 3 µl protein samples to glow-discharged (PELCO easiGlow Ted Pella Inc, at 15 mA for 1 min) holey carbon grids (Quantifoil R 1.2/1.3, Micro Tools GmbH, Germany). The grids were blotted for 4 s and vitrified by rapidly plunging into liquid ethane at −182°C using Leica EM GP plunger (Leica Microsystems, Vienna, Austria). The frozen grids were stored in liquid nitrogen until the day of cryo-EM data collection.

## Cryo-EM data acquisition and processing

In this paper we present data that were collected with three separate cryo-electron microscopes:

1. F30 Polara in Ben-Gurion University, Israel was used for all sample preparation optimization. It was also used to collect data sets without and with potential inhibitors (shown in *Figure 1C* and the ATP and NMN supplemented classes in *Figure 4B*). Finally, it was used for data collection and the 3D reconstructions presented in *Figure 1D*.

    Samples were imaged under low-dose conditions on a FEI Tecnai F30 Polara microscope (FEI, Eindhoven) operating at 300 kV. Datasets were collected using SerialEM

(*Mastronarde, 2005*) on a K2 Summit direct electron detector fitted behind an energy filter (Gatan Quantum GIF) with a calibrated pixel size of 1.1 Å. The energy filter was set to remove electrons >±10 eV from the zero-loss peak energy. The defocus range was set from −1.0 μm to −2.5 μm. The K2 summit camera was operated in counting mode at a dose rate of 8 electrons/pixel/s on the camera. Each movie was dose fractionated into 50 image frames, with total electron dose of 80ē/Å$^2$. Dose-fractionated image stacks were aligned using MotionCorr2 (*Li et al., 2013*), and their defocus values estimated by Gctf (*Zhang, 2016*). The sum of the aligned frames was used for further processing and the rest of the processing was done in Cryosparc V2 (*Punjani et al., 2017*). Particles were auto-picked and subjected to local motion correction to correct for beam-induced drift and then 2D classification with 50 classes. The best (based on shape, number of particles, and resolution) classes were manually selected containing 5459 (for hSARM1$^{E642Q}$) and 43868 (for SAM$^{1-2}$) particles. One initial 3D reference was prepared from all particles and 3D refinement imposing C8 symmetry resulted in the final map.

2. Titan Krios in ESRF CM01 beamline (*Kandiah et al., 2019*) at Grenoble, France, was used for data collection and 3D reconstruction of the GraFix-ed (*Figures 2* and *3*) and NAD+ supplemented (*Figure 5*, *Figure 5—figure supplement 1*) samples.

   Frozen grids were loaded onto a 300kV Titan Krios (ThermoFisher) electron microscope (CM01 beamline at ESRF) equipped with a K2 Summit direct electron-counting camera and a GIF Quantum energy filter (Gatan). Cryo-EM data were acquired with EPU software (FEI) at a nominal magnification of ×165,000, with a pixel size of 0.827 Å. The grid of the GraFix-ed sample was collected in two separate sessions. The movies were acquired for 7 s in counting mode at a flux of 7.06 electrons per Å2 s–1 (data collection 1: 3748 movies) or 6.83 electrons per Å2 s–1 (data collection 2: 4070 movies), giving a total exposure of ~50 electrons per Å2 and fractioned into 40 frames. A total of 7302 movies of the NAD+ supplemented grid sample were acquired for 4 s in counting mode at a flux of 7.415 electrons per Å2 s–1, giving a total exposure of ~40 electrons per Å2 and fractioned into 40 frames. For each data collection a defocus range from −0.8 μm to −2.8 μm was used. Using the SCIPION wrapper (*Martínez et al., 2020*) the imported movies were drift-corrected using MotionCor2 and CTF parameters were estimated using Gctf for real-time evaluation. Further data processing was conducted using the cryoSPARC suite. Movies were motion-corrected and contrast transfer functions were fitted. Templates for auto-picking were generated by 2D classification of auto picked particles. For the GraFix-ed data, template-based auto-picking produced a total of 658,575 particles, from which 147,232 were selected based on iterative reference-free 2D classifications for reconstruction of the GraFix-ed structure. In the case of the NAD+ supplemented data, a total of 335,526 particles were initially picked, from which 159,340 were selected based on iterative reference-free 2D classifications for reconstruction. Initial maps of both GraFix-ed and NAD+ supplemented hSARM1 were calculated using ab initio reconstruction and high-resolution maps were obtained by imposing C8-symmetry in nonuniform 3D refinement. Working maps were locally filtered based on local resolution estimates.

3. Talos Glacios in EMBL, Grenoble, France was used for data collection and the comparison of NAD+ supplemented and not-supplemented samples (*Figure 4A and B*).

   Frozen grids were loaded onto a 200kV Talos Glacios (ThermoFisher) electron microscope equipped with a Falcon3 direct electron-counting camera (ThermoFisher). Cryo-EM data were acquired with EPU software (FEI) at a nominal magnification of ×120,000, with a pixel size of 1.224 Å. For the comparative analysis of NAD+ supplement, the grids of +5mM NAD and no NAD sample were screened and collected. The movies were acquired for 1.99 s in linear mode at a flux of 21.85 electrons per Å2 s–1 (data collection +5mM NAD: 2408 movies; no NAD: 2439 movies) giving a total exposure of ~44 electrons per Å2 and fractioned into 40 frames. For each data collection a defocus range from −0.8 μm to −2.8 μm was used. Warp (*Tegunov and Cramer, 2019*) was used for real-time evaluation, for global and local motion correction, and estimation of the local defocus. The deep learning model within Warp detected particles sufficiently. Inverted and normalized particles were extracted with a boxsize of 320 pixels. A total of 466,135 particles of the +5 mM NAD data and 414,633 particles of the 'no NAD+' set were imported into Cryosparc for further processing and subjected to a 2D circular masked classification with 100 classes. The class averages were manually evaluated and designated as either 'full ring' or 'core ring'.

## Cell viability assay

HEK293T cells were seeded onto lysine precoated 24-well plates (100,000 cells in each well) in a final volume of 500 µl of DMEM (10% FBS) and incubated overnight at 37°C under 5% $CO_2$. They were then transfected with different hSARM1 constructs using the calcium phosphate-mediated transfection protocol (*Kingston et al., 2003*), with addition of 25 µM chloroquine (SIGMA) right before the transfection. Six hours after transfection, the chloroquine-containing DMEM was replaced by fresh complete medium. After 24 hr the medium was removed and replaced with 0.03 mg/ml Resazurin sodium salt (SIGMA) dissolved in Phenol Red-free DMEM. All plates were then incubated for 1 hr at 37°C and measured using a SynergyHI (BioTek) plate reader at 560 nm excitation and 590 nm emission wavelengths. All fluorescent emission readings were averaged and normalized by subtracting the Resazurin background (measured in wells without cells) and then divided by the mean fluorescence emission from cells transfected by the empty vector (pCDNA3).

HEK293F cells were seeded in 24-well plates (1 million cells in each well) in a final volume of 1 ml of FreeStyleTM 293 medium (GIBCO). The cells were transfected with 1 µg DNA as described before and incubated at 37°C and in 8% $CO_2$. Live cells were counted using the trypan blue viability assay every 24 hr for 3 days. Three repeats were performed for each construct.

Noteworthy, the identity of the HEK293F and HEK293T cell lines is clear by their different growth conditions and features.

## In vitro hSARM1 NADase activity assays

For quantitation of hSARM1 NADase activity and the inhibitory effect of selected compounds (*Figure 3F*; *Figure 4A and D*), purified hSARM1[w.t.] and hSARM1E642Q proteins were first diluted to 400 nM concentration in 25 mM HEPES pH 7.5, 150mM NaCl, and then mixed in 25°C with 1 µM of NAD+ (in the same buffer) with a 1:1 v/v ratio. All inhibitors were diluted with the same buffer, and the pH values were measured and if necessary titrated to 7.5. Inhibitors were pre-incubated for 20 min with hSARM1 in 25°C before mixing with NAD+. At designated time points, reactions were quenched by placing the reaction tubes in 95°C for 2 min.

Measurement of NAD+ concentrations was made by a modified enzymatic coupled cycling assay (*Kanamori et al., 2018*). The reaction mix, which includes 100 mM phosphate buffer pH=8, 0.78% ethanol, 4 µM FMN (Riboflavin 5'-monophosphate sodium salt hydrate), 27.2 U/ml Alcohol dehydrogenase (SIGMA), 1.8 U/ml Diaphorase (SIGMA), and freshly dissolved (in DDW) 8 µM resazurin (SIGMA), was added to each sample at 1:1 (v/v) ratio and then transferred to a 384-well black plate (Corning). Fluorescent data were measured using a SynergyHI (BioTek) plate reader at 554-nm excitation and 593-nm emission wavelengths. Standard curve equation for calculation of NAD+ concentration was created for each assay from constant NAD+ concentrations.

### eNAD-based NADase assay

eNAD (nicotinamide 1,N6-ethenoadenine dinucleotide, SIGMA – N2630) was solubilized in water and mixed with native NAD+ in a ratio of 1:10 (mol:mol) to a final stock concentration of 10 mM. Serial dilutions were made with 25 mM HEPES pH 7.5, 150 mM NaCl buffer, and the final mix was transferred to a 384-well black plate (Corning). Reaction started by the addition of hSARM1 to a final concentration 400 nM. Then, eNAD degradation rate was monitored by fluorescence reading (330-nm excitation and 405-nm emission wavelengths) of the plate using a SynergyHI (BioTek) in 25°C for 3 hr. For each NAD+ concentration, a control reaction without SARM1 was measured and subtracted from the +hSARM1 reading and the slope of the linear area was calculated. For the final plot, average of slopes from three separate assays for each concentration was calculated.

### HPLC analysis

Purified hSARM1[w.t.] was first diluted to 800 nM in 25 mM HEPES pH 7.5, 150mM NaCl, and then mixed at 37°C with different concentrations of NAD+ (in the same buffer) in a 1:1 v/v ratio and incubated for 0, 5, and 30 min. 1:100 (v/v) BSA (NEB Inc, 20 mg/ml) was included, and reactions were stopped by heating at 95°C for 2 min. Where specified, NMN (Sigma-Aldrich - N3501) was added in different concentrations. For control, NAD+ consumption was compared to a commercially available porcine brain NADase 0.025 units/ml (Sigma-Aldrich – N9879). HPLC measurements were performed using a Merck Hitachi Elite LaChrom HPLC system equipped with an autosampler, UV detector, and

quaternary pump. HPLC traces were monitored at 260 nm and integrated using EZChrom Elite software. Ten microliters of each sample were injected onto a Waters Spherisorb ODS1 C18 RP HPLC Column (5 μm particle size, 4.6 mm × 150 mm ID). HPLC solvents are as follows: A: 100% methanol; B: 120mM sodium phosphate pH 6.0; C double-distilled water (DDW). The column was pre-equilibrated with B:C mixture ratio of 80:20. Chromatography was performed at room temperature with a flow rate of 1.5 ml/min. Each analysis cycle was 12 min long as follows (A:B:C, v/v): fixed 0:80:20 from 0 to 4 min; gradient to 20:80:0 from 4 to –6 min; fixed 20:80:0 from 6 to 9 min, gradient to 0:80:20 from 9 to 10 min; fixed 0:80:20 from 10 to 12 min. The NAD+ hydrolysis product ADPR was eluted at the isocratic stage of the chromatography while NAD+ elutes in the methanol gradient stage.

## Calculation of SARM1 kinetic parameters

For $V_{max}$ and $K_m$ determination, the NADase activity assay was performed with several different NAD+ substrate concentrations and sampled in constant time points. For each NAD+ concentration, linear increase zone was taken for slope (V0) calculation. All data were then fitted to the Michaelis–Menten equation using nonlinear curve fit in GraphPad Prism software. $K_{cat}$ was calculated by dividing the $V_{max}$ with protein molar concentration.

## Model building and refinement

### GraFix-ed map

The monomers of known octameric X-ray structures of the hSARM1 SAM[1–2] domains (PDBs 6qwv and 6o0s) were superimposed by the CCP4 (*Winn et al., 2011*) program GESAMT (*Krissinel, 2012*) to identify the conserved regions. The model chosen for MR (molecular replacement) contained single polypeptide residues A406–A546 from the PDB 6qwv (SAM Model). Superposition of all available structures of the hSARM1 TIR domain (PDBs 6o0q, 6o0r, 6o0u, 6o0qv, and 6o1b) indicated different conformations of the protein main chain for the BB loop region (a.a 593–607). Two different MR models were prepared for the hSARM1 TIR domain. TIR Model 1 contained regions 562–592 and 608–700 of the high-resolution structure (PDB 6o1b). TIR Model 2 represented assembly of superimposed polyalanine models of all the available hSARM1 TIR domains. Models were positioned into the density map by MR with use of phase information as implemented in program MOLREP (*Vagin and Teplyakov, 2010*). The shell scripts of the MOLREP EM tutorial were downloaded from https://www.ccpem.ac.uk/docs.php and adapted to allow simultaneous positioning of eight molecular symmetry-related SARM1 copies (MOLREP keyword NCS 800).

The search protocol involved Spherically Averaged Phased Translation Function (SAPTF; MOLREP keyword PRF Y). The recent version of MOLREP (11.7.02; 29.05.2019) uses modification of the original SAPTF protocol (*Vagin and Isupov, 2001*), adapted for work with EM density maps (Alexey Vagin, private communication). It now performs the Phased RF (rotation function) search step in a bounding box of the search model and not in the whole (pseudo) unit cell. Instead of Phased Translation Function step, MOLREP performs Phased RF search at several points in the vicinity of SAPTF peak and, in addition, applies Packing Function to potential solutions.

The SAM Model was positioned into the GraFix-ed density map with a score (Map CC times Packing Function) of 0.753. The positioned SAM model was used as a fixed model in MOLREP when searching for the TIR domain. The TIR Model 1 was positioned with a score of 0.582. The MR search with the TIR Model 2 gave essentially the same solution with a lower overall score of 0.559, but higher contrast. With both SAM[1–2] and TIR domains' positions fixed, the MR search for an ideal 10-residue α-helical model allowed location of 64 fragments (eight helices per SARM1 monomer) with scores in the range of 0.78–0.81. These helical fragments were used for building of the ARM domain in Coot (*Emsley et al., 2010*). The quality of the high resolution GraFix-ed density map was sufficient for assignment of side chains for all ARM domain residues. The TIR domain BB loop region (a.a 595–607) was built to fit a relatively poor density map in a conformation different to those observed in X-ray structures. The model was refined using REFMAC5 (*Murshudov et al., 2011*). Side chains of some Lys residues had blobs of undescribed density attached to them. These were modeled as glutaraldehyde ligands. Geometrical restraints for the di-glutaraldehyde molecule and its links to side chains of Lys residues were prepared using JLIGAND (*Lebedev et al., 2012*). The dictionary file was manually edited to allow links to more than a single lysine residue.

## NAD-supplemented map

Originally, the refined full-length GraFix-ed model was positioned by MR into the NAD-supplemented density map, but differences in the relative positions of the hSARM1 domains were apparent. Therefore, MR search was conducted for separate hSARM1 domains. The SAM model was positioned with a score of 0.653 into this map. With the fixed SAM model the TIR Model 1 was found with a score of 0.594. With fixed SAM and TIR Models, the ARM domain from the GraFix model was found with a score of 0.623. The BB-loop of the TIR domain was built into well-defined density map in a conformation not observed in any of the X-ray structures and different to that in the GraFix-ed model. A low sigma cutoff map allowed modeling of the loop connecting the SAM and TIR domains. Inspection of the maps indicated NAD+ binding accompanied by structural rearrangement of the ARM1-ARM2 linker region (a.a. 312–324).

## Accession numbers

Coordinates and structure factors have been deposited in the Protein Data Bank with accession numbers 6ZFX, 7ANW, 6ZG0, 6ZG1, and in the EMDB with accession numbers 11187, 11834, 11190, 11191 for the GraFix-ed, NAD+ supplemented, not treated, and SAM[1–2] models and maps, respectively.

## Acknowledgements

We thank the staff of beamline CM01 of ESRF and members of the Opatowsky lab for technical assistance. We thank Matan Avivi for technical help with HPLC and Gershon Kunin for IT management.

This work was supported by funds from ISF grants no. 1425/15 and 909/19 to YO.

AY is an incumbent of the Jack and Simon Djanogly Professorial Chair in Biochemistry.

## Additional information

### Funding

| Funder | Grant reference number | Author |
| --- | --- | --- |
| Israel Science Foundation | 1425/15 | Yarden Opatowsky |
| Israel Science Foundation | 909/19 | Yarden Opatowsky |
| Elsevier Foundation | Jack and Simon Djanogly Professorial Chair in Biochemistry | Avraham Yaron |

I declare that the funders had no role in study design, data collection and interpretation, or the decision to submit the work for publication.

### Author contributions

Michael Sporny, Formal analysis, Investigation, Methodology; Julia Guez-Haddad, Data curation, Formal analysis, Supervision, Investigation, Methodology, Project administration, Writing - review and editing; Tami Khazma, Data curation, Formal analysis, Investigation, Methodology; Avraham Yaron, Conceptualization, Writing - review and editing; Moshe Dessau, Data curation, Writing - review and editing; Yoel Shkolnisky, Methodology; Carsten Mim, Data curation, Methodology, Writing - review and editing; Michail N Isupov, Formal analysis, Validation, Investigation, Methodology, Writing - review and editing; Ran Zalk, Data curation, Formal analysis, Methodology; Michael Hons, Data curation, Formal analysis, Validation, Methodology, Writing - review and editing; Yarden Opatowsky, Conceptualization, Resources, Formal analysis, Supervision, Funding acquisition, Validation, Investigation, Visualization, Methodology, Writing - original draft, Project administration, Writing - review and editing

## Author ORCIDs
Avraham Yaron (iD) http://orcid.org/0000-0001-9340-7245
Moshe Dessau (iD) http://orcid.org/0000-0002-1954-3625
Carsten Mim (iD) https://orcid.org/0000-0001-6402-8270
Yarden Opatowsky (iD) https://orcid.org/0000-0002-9609-1204

## Decision letter and Author response
Decision letter https://doi.org/10.7554/eLife.62021.sa1
Author response https://doi.org/10.7554/eLife.62021.sa2

# Additional files
## Supplementary files
• Transparent reporting form

## Data availability
Coordinates and structure factors have been deposited in the Protein Data Bank with accession numbers 6ZFX, 7ANW, 6ZG0, 6ZG1, and in the EMDB with accession numbers 11187, 11834, 11190, 11191 for the GraFix-ed, NAD+ supplemented, not treated, and SAM1-2 models and maps, respectively.

The following datasets were generated:

| Author(s) | Year | Dataset title | Dataset URL | Database and Identifier |
|---|---|---|---|---|
| Sporny M, Guez-Haddad J, Khazma T, Yaron A, Mim C, Isupov MN, Zalk R, Dessau M, Hons M, Opatowsky Y | 2020 | hSARM1 GraFix-ed | https://www.rcsb.org/structure/6ZFX | RCSB Protein Data Bank, 6ZFX |
| Sporny M, Guez-Haddad J, Khazma T, Yaron A, Mim C, Isupov MN, Zalk R, Dessau M, Hons M, Opatowsky Y | 2020 | hSARM1 NAD+ complex | https://www.rcsb.org/structure/7ANW | RCSB Protein Data Bank, 7ANW |
| Sporny M, Guez-Haddad J, Khazma T, Yaron A, Mim C, Isupov MN, Zalk R, Dessau M, Hons M, Opatowsky Y | 2020 | SARM1 SAM1-2 domains | https://www.rcsb.org/structure/6ZG0 | RCSB Protein Data Bank, 6ZG0 |
| Sporny M, Guez-Haddad J, Khazma T, Yaron A, Mim C, Isupov MN, Zalk R, Dessau M, Hons M, Opatowsky Y | 2020 | SARM1 SAM1-2 domains | https://www.rcsb.org/structure/6ZG1 | RCSB Protein Data Bank, 6ZG1 |
| Sporny M, Guez-Haddad J, Khazma T, Yaron A, Mim C, Isupov MN, Zalk R, Dessau M, Hons M, Opatowsky Y | 2020 | hSARM1 NAD+ complex | http://emsearch.rutgers.edu/atlas/11834_summary.html | EMDataBank, 11834 |
| Sporny M, Guez-Haddad J, Khazma T, Yaron A, Mim C, Isupov MN, Zalk R, Dessau M, Hons M, Opatowsky Y | 2020 | hSARM1 GraFix-ed | http://emsearch.rutgers.edu/atlas/11187_summary.html | EMDataBank, 11187 |
| Sporny M, Guez-Haddad J, Khazma | 2020 | SARM1 SAM1-2 domains | http://emsearch.rutgers.edu/atlas/11190_sum- | EMDataBank, 11190 |

| T, Yaron A, Dessau M, Mim C, Isupov MN, Zalk R, Hons M, Opatowsky Y | | | | mary.html | |
| Sporny M, Guez-Haddad J, Khazma T, Yaron A, Dessau M, Mim C, Isupov MN, Zalk R, Hons M, Opatowsky Y | 2020 | SARM1 SAM1-2 domains | http://emsearch.rutgers.edu/atlas/11191_summary.html | EMDataBank, 11191 |

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
