## [Decision Letter]

**Acceptance summary:**

The manuscript reports the unexpected finding that the NADase SARM1, a mediator of axon loss in response to injury, is inhibited by binding of NAD+ substrate to an allosteric site. NAD+ is seen to stabilize an ordered conformation of catalytic domains at the periphery of the octameric assembly. This arrangement prevents formation of active catalytic domain dimers, thereby providing an explanation for regulation of the energetic collapse caused by SARM1 activation.

**Decision letter after peer review:**

Thank you for submitting your article "The Structural Basis for SARM1 Inhibition, and Activation Under Energetic Stress" for consideration by *eLife*. Your article has been reviewed by three peer reviewers, including Christopher P Hill as the Reviewing Editor and Reviewer #1, and the evaluation has been overseen by John Kuriyan as the Senior Editor. The following individual involved in review of your submission has agreed to reveal their identity: Jonathan Elegheert (Reviewer #3).

The reviewers have discussed the reviews with one another and the Reviewing Editor has drafted this decision to help you prepare a revised submission.

Summary:

SARM1 is an enzyme that depletes NAD+ to trigger axon loss in response to injury. A recent publication reported the structure of full-length SARM1 to reveal an octameric ring assembly. The current study also reports structures of full-length SARM1 (inactive mutant) by cryo-EM, and makes the additional finding that SARM1 is substrate-inhibited. Structures determined after crosslinking in the presence of glycerol or the presence of NAD+ appear to be an inhibited conformation. Biochemical studies show that glycerol and NAD+ (and ATP) are inhibitors, and NAD+ binds an allosteric site to prevent the catalytic domains adopting their active dimeric association. In contrast, a structure determined in the absence of NAD+ or crosslinking allows enough conformational freedom for catalytic domains to dimerize. Overall, this is an important advance in structural-mechanistic understanding.

Essential revisions:

1) Please discuss the recent Bratkowski et al. publication in Cell Reports from 2020 explicitly, which reports similar structure determination but arrives at a very different mechanistic conclusion.

2) A major point of the paper is on the NAD-mediated allosteric inhibition of SARM1. To strengthen this point, it may be helpful to expand the discussion and include more figure panels that illustrate how NAD binds SARM1, and how this binding stabilizes the autoinhibited conformation. Given the limitations of data, the best way to illustrate and discuss will be a judgement call for the authors. We note that the density for NAD as shown in Figure 5 is a bit weak, with a part of the NAD molecule sticking outside of the density. Is it possible to fit NAD in the opposite orientation? In addition, would it be helpful to show comparison the ARM/TIR interface with or without NAD in more detail, which may shed more light on how the NAD-induced conformation of ARM holds tighter to TIR.

3) As pointed out the manuscript, the results from mutations at the NAD binding site are somewhat unexpected. L152A, which has a strong effect, appears not directly contacting NAD. R322 and R157 also seem a little far from NAD. Plus, these residues are not conserved in *C. elegans* SARM1. In contrast, W103 is conserved and appears to be making a key interaction with NAD, but its mutation has no effect. Related to this point, the recent Cell Reports manuscript shows essentially the same autoinhibited assembly of SARM1, in the absence of Grafix crosslinking or NAD, which seems to argue that the SARM1 is able to adopt the autoinhibited conformation on its own. These discrepancies raise the possibility that in cells a different metabolite binds to this site in SARM1, with similar but distinct atomic interactions. Or is it possible that the NAD-mediate regulation is species specific?

4) The secondary TIR docking site is disengaged in the NAD-bound structure. Can the authors discuss whether this reflects the actual difference between the two states of the protein, or is it possible that the secondary docking site is imposed by the crosslinking reagent?

5) The resolution based on the FSC between the density map and model for the NAD-bound structure is 5.6 Å, much worse than the resolution of the map based on the gold-standard FSC (2.7). This large difference between the two FSC curves is concerning, suggesting that there are some issues in the model that cause it to not fit the density well. This issue needs to be addressed, and the PDB-vs-map FSC curves should be shown as supplemental figures. The manuscript does touch on this point, mentioning "domain-based heterogeneity", but fell short of providing a clear explanation. One possibility might be that NAD only occupied some, but not all, of the subunits in the octamer, which resulted in different conformations of the peripheral TIR and ARM domains within the same octamer. Imposing C8 symmetry under this situation would deteriorate the density for all the subunits. Have the authors tried the symmetry expansion and then focused refinement approach as implemented in Relion (http://dx.doi.org/10.1016/bs.mie.2016.04.012)? One might then perform classification of the subunit, which may allow subunits with different conformations to be separated.

6) The HEK-based viability assay has been used in previous studies to assess SARM1 activity and the effect of various mutations and domain deletions. Hence, the authors adopted it for this work. Although the experimental setup is internally controlled (accounting for fluorescent background signals and emission from mock-transfected cells), there is no control over or validation of mutant protein expression levels (assessed using e.g. blot densitometry), which may skew the cell viability data if expression levels would diverge from wild-type levels, by impacting on NAD depletion rates. Overall, the effects of various SARM1 mutants would be more elegantly assessed using a robust neuronal assay, where SARM1 variants would be expressed in a more native-like environment and axonal death would be monitored. Although desirable, this is not necessary for the revised manuscript.

---

## [Author Response]

Essential revisions:1) Please discuss the recent Bratkowski et al. publication in Cell Reports from 2020 explicitly, which reports similar structure determination but arrives at a very different mechanistic conclusion.

Bratkowski et al. is now discussed in the context of structural comparison at the end of the “2.8-6.5 Å resolution structure….” section, and for the activation model by increased NMN vs. decreased NAD+ at the “Conclusions…” section.

2) A major point of the paper is on the NAD-mediated allosteric inhibition of SARM1. To strengthen this point, it may be helpful to expand the discussion and include more figure panels that illustrate how NAD binds SARM1, and how this binding stabilizes the autoinhibited conformation. Given the limitations of data, the best way to illustrate and discuss will be a judgement call for the authors.

2.1) We have now added a new figure (Figure 5—figure supplement 2) to help follow the NAD+ allosteric binding site and conformational changes there.

The issue of “how this binding stabilizes the autoinhibited conformation” is still not entirely clear. Based on the comparison between the structures of the NAD+ supplemented SARM1 to the GraFixed NAD+ free structure, we see that NAD+ binds at the ARM allosteric inhibitory site and causes some structural changes there (as detailed in the paper and the new Figure 5—figure supplement 2). However, it is not clear how and if these limited changes induce the compact inhibited two-ring conformation. It is important to keep in mind that the GraFixed structure (as well as the one published in Cell Reports by Bratkowski et al.) do not represent active SARM1, where the outer ring is disassembled. Rather, these structures represent a small sub-population of particles that retain the two-ring inhibited conformation even in the absence of NAD+. Possibly, the structure of the ARM domain in the disassembled conformation (which is currently not available) is such that prevents the assembly of the outer ring, and that NAD+ binding to it induces a dramatic structural change that allows the outer ring assembly.

We note that the density for NAD as shown in Figure 5 is a bit weak, with a part of the NAD molecule sticking outside of the density.

2.2) Most of the NAD+ density is as strong as the surrounding polypeptide elements. There is a very weak density for the adenosine, but the rest of the molecule is very clear, including the positions of the phosphate oxygens, and the amide protrusion on the nicotine ring.

Is it possible to fit NAD in the opposite orientation?

2.3) When positioned opposite to the current orientation, the adenosine sticks out of the (nicotinamide) density. In numerical terms, the map/model CC per residue (by phenix calculation) for the NAD+ drops from 0.88 to 0.85. More importantly – the adenosine clashes with surrounding sidechains, which have a very clear density themselves. Therefore, the opposite orientation of the NAD+ is unlikely.

In addition, would it be helpful to show comparison the ARM/TIR interface with or without NAD in more detail, which may shed more light on how the NAD-induced conformation of ARM holds tighter to TIR.

2.4) As explained above (2.1), how NAD+ binding stabilizes the compact two-ring conformation is still not clear, probably because of the absence of ARM domain structure in the disassembled conformation. It is clear that the NAD+ binding site is inhibitory – because mutations at this site activate SARM1 in cultured cells.

There are differences between the GraFix-ed and NAD+ supplemented structures at the TIR domain’s BB loop interaction with the secondary ARM site. Alas, the BB loop (which is one of the most important elements in SARM1) is one of the least clear regions in both density maps, and therefore an in-depth analysis of the changes in the BB loop may be speculative.

3) As pointed out the manuscript, the results from mutations at the NAD binding site are somewhat unexpected. L152A, which has a strong effect, appears not directly contacting NAD.

3.1) The backbone amine of L152 forms a hydrogen bond with one of the NAD+ ribose oxygens (see the new 2D interaction map Figure 5—figure supplement 2). More importantly, the L152 sidechain is positioned in a way that may affect the conformation of the 𝛂5-6 turn, and therefore the alanine mutation should inflict a conformational effect in this region. The fact that the L152A strongly activates SARM1 best reflects, in our opinion, the critical role of the ARM^1^-ARM^2^ “crescent horns” region in regulating SARM1 inhibition and activation.

R322 and R157 also seem a little far from NAD.

3.2) R157 forms a salt bridge with a NAD+ phosphate (see the new 2D interaction map Figure 5—figure supplement 2). R322 is involved in hydrophobic interactions with NAD+ atoms, and more importantly the R322 sidechain stabilizes the ARM^1^-ARM^2^ loop, and therefore the glutamate mutation should destabilize this region, explaining the activating effect of this mutation.

Plus, these residues are not conserved in C. elegans SARM1.

3.3) It is possible that not all the mechanisms that control SARM1 activation and inhibition are conserved in *C. elegans*, and it acquired additional regulations during evolution. It should be pointed that Wallerian degeneration is not conserved in *C. elegans*, as axonal degeneration is induced only by two nerve cuts, where there is a mild protective effect (~20%) of TIR-1/SARM1 loss (doi: https://doi.org/10.1101/2020.06.23.165852). This is in a sharp contrast to the observations in flies and mice.

In contrast, W103 is conserved and appears to be making a key interaction with NAD, but its mutation has no effect.

3.4) The ineffectiveness of the W103A mutation troubled us too, and we were able to generate and test another mutation: W103D that indeed, as expected, induced strong activation of SARM1. We think that while the substitution to alanine deprived a hydrophobic contact point, which could be compensated by the other interactions, the interference made by the aspartate substitution could not be tolerated.

Related to this point, the recent Cell Reports manuscript shows essentially the same autoinhibited assembly of SARM1, in the absence of Grafix crosslinking or NAD, which seems to argue that the SARM1 is able to adopt the autoinhibited conformation on its own.

3.5) Only small percentage of purified SARM1 particles adopt the autoinhibited conformation on their own. >80% of SARM1 particles, in the absence of NAD+ in high concentration, are found in the active conformation, where only the inner ring is visible. It can be seen in our Figure 4 B,C, and also in Bratkowski et al. Figure S1B. To overcome this problem, and to determine a cryo-EM structure, Bratowski et al. selected only the minor fraction of the two-ring particles (5.6% of the initial picked particles – see Figure S1F in Cell Reports).

The key difference between the two studies, is that while we have acknowledged that purified SARM1 is mostly active, and therefore focused our studies on finding the inhibitory factor that is lost in purification (NAD+), Bratowski et al. considered the purified SARM1 as inhibited and showed that NMN is an activating factor.

These discrepancies raise the possibility that in cells a different metabolite binds to this site in SARM1, with similar but distinct atomic interactions.

3.6) We, of course, cannot rule out the binding of other cellular elements – small molecules or proteins to the ARM allosteric inhibitory site. For example, being a structural component of NAD+, it is likely that NMN also binds to the ARM^1^ side of this site (see the new addition to the conclusion part in the paper). However, NAD+ is one of the most abundant metabolites in the cell, while NMN concentrations are much lower. So, when we consider together the NAD+ high cellular concentrations, the dramatic effect it has on SARM1 structure, and the clear substrate inhibition curve and values that correspond to the physiological NAD+ concentrations, we are inclined to think that NAD+ is indeed the key metabolite that regulates SARM1 inhibition and activation.

Or is it possible that the NAD-mediate regulation is species specific?

3.7) While SARM1 seems to have an NADase activity in all animals, it is quite clear that there are some differences between species in SARM1 regulation. One such difference between *C. elegans* and other animals is detailed above in answer 3.3. Another difference is the presence of an N-terminal mitochondrial localization peptide in mammals – but not in non-mammalian creatures (based on our analysis).

4) The secondary TIR docking site is disengaged in the NAD-bound structure. Can the authors discuss whether this reflects the actual difference between the two states of the protein, or is it possible that the secondary docking site is imposed by the crosslinking reagent?

As we have discussed above in 2.4, there are differences at the TIR domain’s BB loop interaction with the secondary ARM site. Alas, the BB loop (which is one of the most important elements in SARM1) is one of the least clear regions in the structure, and therefore an in-depth analysis of the changes in the BB loop may be speculative.

5) The resolution based on the FSC between the density map and model for the NAD-bound structure is 5.6 Å, much worse than the resolution of the map based on the gold-standard FSC (2.7). This large difference between the two FSC curves is concerning, suggesting that there are some issues in the model that cause it to not fit the density well. This issue needs to be addressed, and the PDB-vs-map FSC curves should be shown as supplemental figures. The manuscript does touch on this point, mentioning "domain-based heterogeneity", but fell short of providing a clear explanation. One possibility might be that NAD only occupied some, but not all, of the subunits in the octamer, which resulted in different conformations of the peripheral TIR and ARM domains within the same octamer. Imposing C8 symmetry under this situation would deteriorate the density for all the subunits. Have the authors tried the symmetry expansion and then focused refinement approach as implemented in Relion (http://dx.doi.org/10.1016/bs.mie.2016.04.012)? One might then perform classification of the subunit, which may allow subunits with different conformations to be separated.

We apologize for that – some of the numbers in the submitted table were old, and the model has much better correlation figures. As we have explained above (2.2) most of the NAD+ density is actually as strong as the surrounding polypeptide elements, and we see no indication that the NAD+ binding is not homogenous.

When we do not impose C8 symmetry during the NAD-supplemented structure processing, we do not see differences in NAD+ occupancy between protomers. Also, masking of protomers and symmetry expansion followed by 3D classification yields maps with low resolution that falls short of revealing different species in terms of NAD+.

6) The HEK-based viability assay has been used in previous studies to assess SARM1 activity and the effect of various mutations and domain deletions. Hence, the authors adopted it for this work.

6.1) We are the first to introduce a suspension culture of HEK293F for SARM1 viability assay. We found the suspension culture assay to be simple and most reproducible, and therefore the best way to monitor SARM1 NADase activity in cultured cells.

Although the experimental setup is internally controlled (accounting for fluorescent background signals and emission from mock-transfected cells), there is no control over or validation of mutant protein expression levels (assessed using e.g. blot densitometry), which may skew the cell viability data if expression levels would diverge from wild-type levels, by impacting on NAD depletion rates.

6.2) The basic problem is that constitutively active SARM1 truncations and mutants rapidly kill the cells, and we observe cell death as soon as 8 hours post transfection in some of the mutants. So, clearly WB will not be feasible as the protein levels are dropping due to cell death.

Overall, the effects of various SARM1 mutants would be more elegantly assessed using a robust neuronal assay, where SARM1 variants would be expressed in a more native-like environment and axonal death would be monitored. Although desirable, this is not necessary for the revised manuscript.

Due to the Covid-19 lockdown we are unable to run experiments with neurons. Thank you for the consideration.